# An Empirical Study of Deep Reinforcement Learning in Continuing Tasks

## Abstract

In reinforcement learning (RL), continuing tasks refer to tasks where the agent-environment interaction is ongoing and can not be broken down into episodes. These tasks are suitable when environment resets are unavailable, agent-controlled, or predefined but where all rewards—including those beyond resets—are critical. These scenarios frequently occur in real-world applications and can not be modeled by episodic tasks. While modern deep RL algorithms have been extensively studied and well understood in episodic tasks, their behavior in continuing tasks remains underexplored. To address this gap, we provide an empirical study of several well-known deep RL algorithms using a suite of continuing task testbeds based on Mujoco and Atari environments, highlighting several key insights concerning continuing tasks. Using these testbeds, we also investigate the effectiveness of a method for improving temporal-difference-based reinforcement learning (RL) algorithms in continuing tasks by centering rewards, as introduced by Naik et al. (2024). While their work primarily focused on this method in conjunction with Q-learning, our results extend their findings by demonstrating that this method is effective across a broader range of algorithms, scales to larger tasks, and outperforms two other reward-centering approaches.

## 1 Introduction

Reinforcement learning (RL) tasks can generally be divided into two categories: episodic tasks and continuing tasks. In episodic tasks, the interaction between the agent and environment naturally breaks down into distinct episodes, with the environment resetting to an initial state at the end of each episode. The goal of these tasks is to maximize the expected cumulative reward within each episode. Episodic tasks are suitable when the environment can be reset, the reset conditions are predefined, and rewards beyond the reset point do not matter—such as in video games.

In contrast, continuing tasks involve ongoing agent-environment interactions where all rewards matter. Continuing tasks are well-suited for situations where the environment cannot be reset. In many real-world problems, such as inventory management, content recommendation, and portfolio management, the environment's dynamics are beyond the control of the solution designer, making environment resets impossible. Continuing tasks can also be useful when resets are possible. First, when designing reset conditions is challenging, it can be beneficial for the agent to determine when to reset. For instance, a house-cleaning robot might decide to reset its environment by requesting to be placed back on the charging dock if trapped by cables. The second scenario involves predefined reset conditions, just as in episodic tasks, but where post-reset rewards still matter. For example, when training a robot to walk, allowing the robot to learn when to fall and reset can lead to better overall performance, as it could pursue higher rewards after resetting rather than merely avoiding falling at all costs. In both scenarios, continuing tasks provide an opportunity to balance the frequency of resets and the rewards earned by choosing the cost of reset, which is a flexibility not present in episodic tasks.

Continuing tasks can also be useful in cases where the ultimate goal is to solve an episodic task. This is best exemplified by the works on the autonomous RL setting, where the goal is to address an episodic task, and the agent learns a policy to reset the environment. In this setting, the agent is trained on a special continuing task, where the main task, which is the episodic task of interest, and an auxiliary task, such as moving to the initial state, are presented in an interleaved sequence. The

learned main task's policy is deployed after training. This setting can be most useful when resets are expensive, and it is possible to reach the initial state from all other states, such as in many robotic tasks. Representative works in this direction include Eysenbach et al. (2017); Sharma et al. (2021); Zhu et al. (2020) and Sharma et al. (2022).

Despite the broad applications of continuing tasks, empirical studies on deep RL algorithms in these tasks remain limited, and their unique challenges remain under-explored. Most existing empirical studies focus on demonstrating better performance of new algorithms. For instance, Zhang and Ross (2021), Ma et al. (2021), Saxena et al. (2023), and Hisaki and Ono (2024) introduced average-reward variations of popular deep RL algorithms and empirically evaluated them alongside their discounted return counterparts on continuing tasks based on the Mujoco environment (Todorov et al., 2012), highlighting improvements in performance. In addition to the Mujoco testbeds used in the above works, Platanios et al. (2020) and Zhao et al. (2022) provided new testbeds for continuing tasks. However, Platanios et al.'s (2020) testbed also presents significant partial observability, making it not suitable for isolating the challenges of continuing tasks. The testbeds presented by Zhao et al. (2022) have small discrete state and action spaces, making them primarily suitable for studying tabular algorithms. To our knowledge, only two empirical studies have explored the unique challenges that continuing tasks present to deep RL algorithms. In particular, Sharma et al. (2022) found that several RL algorithms designed for the autonomous RL setting perform significantly worse when resets are unavailable. This indicates that resets limit the range of visited states, focusing the agents around initial and goal states. Naik et al. (2024) demonstrated that in two small-scale continuing tasks (namely, Pendulum and Catch), the DQN algorithm performs poorly when using a large discount factor or when rewards share a common offset. While a large discount factor also poses challenges in episodic tasks, its effects can be masked by the finite length of episodes. Shifting rewards by a common offset can only be applied to continuing tasks, as in episodic tasks, it changes the underlying problem.

Our first contribution is an empirical study of several well-known deep RL algorithms on a suite of continuing task testbeds. The objectives of this study include understanding the challenges present in continuing tasks with different reset scenarios and the extent to which the existing deep RL algorithms address these challenges. The tested algorithms include DDPG (Lillicrap, 2015), TD3 (Fujimoto et al., 2018), SAC (Haarnoja et al., 2018), PPO (Schulman et al., 2017), and DQN (Mnih et al., 2015). The testbeds are obtained by applying simple modifications to existing episodic testbeds from Gymnasium (Towers et al., 2024) based on Mujoco and Atari environments (Bellemare et al., 2013), such as removing time-based resets and treating resets as standard transitions in the environment with some extra cost. We considered the following reset scenarios: no resets, predefined resets, and agent-controlled resets. The proposed testbeds include 15 continuous action tasks covering all these reset scenarios and six discrete action tasks with predefined resets. We did not create Atari-based testbeds without resets or with agent-controlled resets because it is not trivial to remove the predefined resets there. While some of our Mujoco testbeds are identical to those used in prior works studying average-reward algorithms (e.g., Zhang and Ross 2021), the majority differ from theirs. The code used in this study is based on the Pearl library (Zhu et al., 2023) and will be available upon the publication of this paper.

The empirical study reveals several key insights. First, the tested algorithms perform significantly worse in tasks without resets compared to those with predefined resets. We found that predefined resets help in at least two ways. One is that they limit the effective state space the agent needs to deal with. This point echoes Sharma et al.'s (2022) finding in the autonomous RL setting. The other way is that they move the agent back to an initial state when the agent fails to escape from suboptimal states due to the weak exploration ability. Second, tested algorithms in continuing testbeds with predefined resets learn policies outperforming the same algorithms in the episodic testbed variants when both policies are evaluated in the continuing testbeds. We found that better performance is achieved by choosing actions that yield higher rewards at the cost of more frequent resets. Further, increasing the reset cost reduces the number of resets and, interestingly, can even improve overall rewards, indicating that reset costs are not only problem parameters but also solution parameters. Third, when agents are given control over resets, in some cases, it can barely surpass or even be worse than random policies in tasks with predefined resets, which suggests that these tasks are quite challenging for the tested algorithms. Lastly, all algorithms perform poorly in continuing tasks with large discount factors or shared reward offsets, which is in line with Naik et al.'s (2024) findings about deep Q-learning in small-scale tasks. These findings highlight the need for careful selection

of discount factors and the avoidance of reward offsets when applying these deep RL algorithms to continuing tasks.

Our second contribution is empirically showing the effectiveness of temporal-difference (TD)–based reward centering on a wide range of deep RL algorithms. Originally proposed by Naik et al. (2024), reward centering is an idea to address challenges posed by a large discount factor and a large common reward offset by subtracting an estimate of the average-reward rate from all rewards. TD-based reward centering is one approach to estimating the reward rate and is particularly beneficial for off-policy algorithms; the reward rate can be estimated using a moving average of past rewards in the on-policy setting but not in the off-policy setting. Naik et al. (2024) demonstrated its effectiveness primarily in the tabular and linear function approximation settings, with deep RL results limited to DQN on two small-scale tasks (Pendulum and Catch) and PPO, which is an on-policy algorithm, on six Mujoco tasks. We show that TD-based reward centering improves all tested algorithms on a larger scale and more diverse testbeds. Additionally, we compare TD-based reward centering with the moving average approach, despite its theoretical issues in the off-policy setting, and an approach using a set of selected reference states (Devraj and Meyn, 2021).

Empirical results demonstrate that TD-based reward centering significantly improves performance across a wide range of continuing tasks and maintains performance in others. Furthermore, algorithms incorporating TD-based reward centering are not sensitive to reward offsets. The findings related to large discount factors present a more nuanced picture compared to Naik et al.'s (2024) results on smaller tasks. While their experiments show that, with reward centering, the discount factor primarily affects the speed of learning without degrading long-term performance even as the discount factor approaches one, our results on larger scale tasks show that long-term performance still declines, albeit much less sharply than when reward centering is not employed. This suggests that even with TD-based reward centering, tuning the discount factor remains valuable, particularly in more complex tasks. Finally, while the moving-average approach is less effective than TD-based reward centering, surprisingly, it is helpful for the tested off-policy algorithms despite its theoretical limitations. The reference-state-based approach improves the tested algorithms in some tasks but hurts in others.

## 2 EVALUATING DEEP RL ALGORITHMS ON CONTINUING TASKS

This section evaluates several of the most well-known RL algorithms in a suite of continuing testbeds.

### 2.1 TESTBEDS WITHOUT RESETS

This section evaluates four continuous control algorithms (DDPG, TD3, SAC, PPO) in five continuing testbeds without resets and shows how the absence of resets poses a significant challenge to the tested algorithms.

The testbeds are based on five Mujoco environments: Swimmer, HumanoidStandup, Reacher, Pusher, and Ant. The goal of the Swimmer and Ant testbeds is to move a controlled robot forward as fast as possible. For Reacher and Pusher, the goal is to control a robot to either reach a target position or push an object to a target position. In HumanoidStandup, the goal is to make a lying Humanoid robot stand up. The episodic versions of these testbeds have been standard in RL (Towers et al., 2024). The continuing testbeds are the same as the episodic ones except for the following differences. First, the continuing testbeds do not involve time-based or state-based resets. For Reacher, we resample the target position every 50 steps while leaving the robot's arm untouched, so that the robot needs to learn to reach a new position every 50 steps. Similarly, for Pusher, everything remains the same except that the object's position is randomly sampled every 100 step. As for Ant, we increase the range of the angles at which its legs can move, so that the ant robot can recover when it flips over.

Note that we created these continuing testbeds based on environments where, except for a set of transient states, it is possible to transition from any state to any other state. This is known as the weakly communicating property in MDPs (Puterman, 2014). Without this property, no algorithm can guarantee the quality of the learned policy because the agent might enter suboptimal states, from which there is no way to escape. An example environment without this property is Mujoco's Hopper, where if the agent falls, it is unable to stand back up.

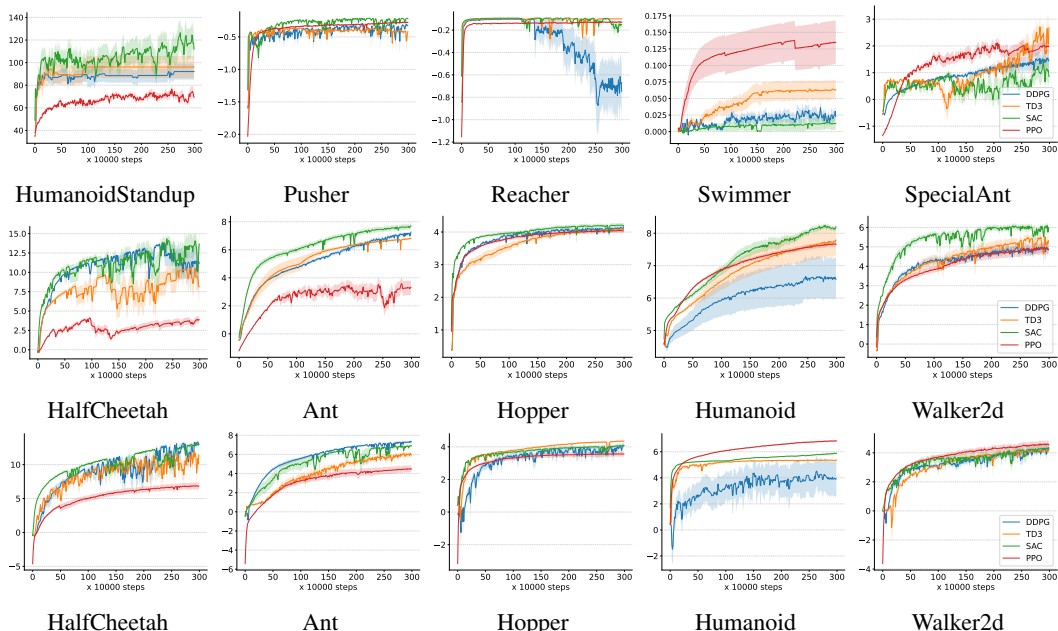

Figure 1: Learning curves in continuing testbeds without resets (upper row), with predefined resets (middle row), and with agent-controlled resets (lower row) based on the Mujoco environment. Each point in a curve shows the reward rate averaged over the past $10,000$ steps. The shading area shows one standard error.

For each task, we ran all tested algorithms for ten independent runs, with each run lasting 3 million steps. The tested parameter settings are provided in Section A.2. We report learning curves corresponding to the parameter setting that results in the highest average-reward rate across the last $10,000$ steps in the upper five plots in Figure 1. We also manually checked the learned policies by rendering videos to see if they performed reasonably well in the tested problems.

For Reacher, we found that TD3 and SAC both learned descent policies in most of the runs, DDPG failed catastrophically after converging to a descent policy in half of the test runs, and

| Task | DDPG | TD3 | SAC | PPO |
|------|------|-----|-----|-----|
| Swimmer | 343.45 | 469.54 | 2428.54 | 29.19 |
| HumanoidStandup | 63.76 | 30.66 | 39.04 | 0.44 |
| Reacher | 394.67 | 0.02 | 10.48 | 3.42 |
| Pusher | -4.10 | 1.65 | -3.30 | 0.67 |
| SpecialAnt | 35.30 | 88.08 | 120.98 | 23.82 |

Table 1: The percentage of the final reward rate improvement when resets are applied with a small probability. The gray color indicates that the performance difference is not statistically significant. This table shows that in some tasks, the lack of resets poses a significant challenge to the tested algorithms.

PPO's learned policies did not reach the target positions across most of the runs. For Pusher, all algorithms learned policies that perform reasonably well in most of runs. For Swimmer, Humanoid-Standup, and SpecialAnt, none of the algorithms were able to learn a policy that performed reasonably well in most of the runs.

To understand if the poor performance of the tested algorithms' performance is mainly due to the unavailability of resets, we created three variants of these testbeds where resets occur with probabilities of $0.01, 0.001$, and $0.0001$ per time step, respectively. Upon resetting, regardless of the current state and the chosen action, the resulting next state would be sampled from the task's initial state distribution. The reward setting and the rest of the task dynamics remain unchanged. For each resetting variant, we ran each algorithm for ten runs, each of which consists of 3 million steps. We report the percentage of improvement, defined as $\frac{\bar{r}^{\text{no resets}} - \bar{r}^{\text{random}}}{\bar{r}^{\text{random resets}} - \bar{r}^{\text{random}}} - 1$, where $r^{\text{no resets}}$ is the reward rate of the final policy learned in the task without reset, $r^{\text{random resets}}$ is the best final reward rate across all three variants with resets, and $\bar{r}^{\text{random}}$ is the reward rate of a uniformly random policy in the testbed without resets. All reward rates are averaged over ten runs. We use gray shading to indicate that the

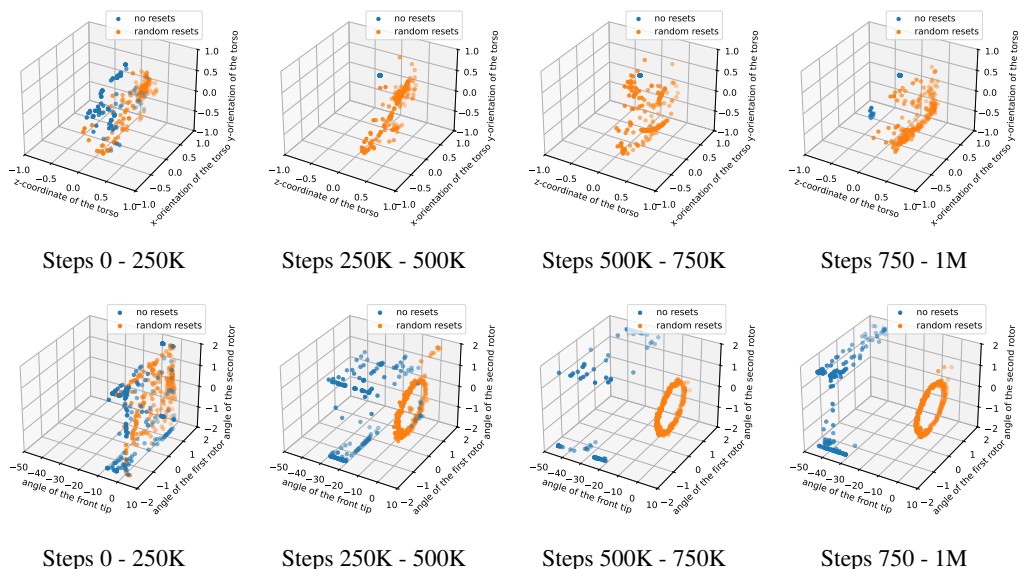

Figure 2: Evolution of DDPG's visited states in two HumanoidStandup testbeds (upper row) and TD3's visited states in two Swimmer testbeds (lower row). In both cases, one testbed does not involve resets, while the other one resets with a probability of 0.001 per time step. We visualize three key elements of the visited states in the first 1M steps of one run. For HumanoidStandup, all blue dots concentrate on a small suboptimal region, indicating that the agent fails to perform a sufficient amount of exploration without resets. For the Swimmer, the orange circle indicates the swimmer undulates like a snake to move forward, suggesting that the agent finds a decent policy. Without resetting, the agent explores a larger region of the state space but fails to learn a good policy.

reward rate difference with and without resets is not statistically significant, as determined by Welch's t-test with a $p$-value less than 0.05. The results (Table 1) show that, overall, the learned policies in the testbeds with random resets are significantly better than those learned in the testbeds without resets.

Visualizing the evolution of some key state elements reveals two reasons why algorithms performed much better in the reset variants of the testbeds. To illustrate these two reasons, we show in a representative run, for every 1000 steps, the evolution of the height and orientation of the Humanoid robot's torso with DDPG and the evolution of the angular component of the Swimmer robot with TD3. In both testbeds with random resets, the reset probability is 0.001. The evolution plots are shown in Figure 2. For HumanoidStandup, the agent's selected state elements concentrate on a point for a long period, suggesting that the agent is trapped in some small region in the state space. Note that the MDP is weakly communicating, therefore it is possible to move from every state to every other state. In addition, note that the $z$-coordinate is the main factor contributing to the task's reward. Hence, a low $z$-value, in general, corresponds to a low reward. Therefore, the evolution plots show that the agent did not perform sufficient exploration to escape from suboptimal states. With random resets, the exploration challenge is significantly simplified because external resets move the agent out of these suboptimal states.

Swimmer's evolution plots show that, as training progresses, the agent eventually discovers a decent policy in the reset variant of the testbed (shown by orange dots). In the original testbed, the algorithm explores a wider range of the state space but fails to converge to an effective policy (shown by blue dots). A closer look at the blue dots reveals that the front tip's angle gradually shifts from 0 to $-50$ rads within the first 1M steps. Notably, there is no inherent limit on how large or small this angle can be, leading the agent to continuously observe novel front tip angles that extrapolate beyond the previously encountered ones and explore ever-larger front tip angles, searching for potentially higher rewards. The testbed variant with resets avoids this challenge by constraining exploration to the vicinity of the initial state, effectively reducing the region the agent could possibly visit in the vast state space.

To verify if the limited size of the state space is indeed the main reason that explains the performance gap in Swimmer, we tested the four algorithms on a variant of the Swimmer testbed with constrained state space. This variant is only different from the original Swimmer in that the angular elements observed by the agent are converted to be within $[-\pi, \pi)$ (i.e., angle $x$ in the original testbed is converted to $x \mod 2\pi - \pi$). Note that this conversion does not change the environment dynamics, and the new state space is equivalent to the original one. We observed that DDPG, TD3, and SAC in this new testbed achieved statistically significantly higher performance compared to the original Swimmer, with the percentage of improvement being 1233.26%, 333.22% and 2287.43 %. For PPO, the performance improvement is not statistically significant. The results show that constraining the state space by resets is indeed a major factor in achieving a higher performance in swimmers with resets and limiting the size of the state space can achieve similar performance gains as resets.

## 2.2 Testbeds with Predefined Resets

This section evaluates both continuous and discrete control algorithms on continuing task testbeds with predefined resets. In addition, it shows how the learned policies differ from policies learned in episodic variants of the testbeds.

The test suite includes both continuous and discrete control testbeds. The continuous control testbeds are built upon five Mujoco environments: HalfCheetah, Ant, Hopper, Humanoid, and Walker2d. In these testbeds, the objective is to control a simulated robot to move forward as quickly as possible. The corresponding existing episodic testbeds involve time-based truncation of the agent's experience followed by an environment reset. In the continuing testbeds, we remove this time-based truncation and reset. However, we retain state-based resets, such as when the robot is about to fall (in Hopper, Humanoid, and Walker2d) or when it flips its body (in Ant). In addition, we add a reset condition for HalfCheetah when it flips, which is not available in the existing episodic testbeds. Each reset incurs a penalty of $-10$ to the reward, punishing the agent for falling or flipping.

The discrete control testbeds are adapted from six Atari environments: Breakout, Pong, Space Invaders, BeamRider, Seaquest, and Ms. PacMan. Like the Mujoco environments, the episodic versions include time-based resets, which we omit in the continuing testbeds. In these Atari environments, the agent has multiple lives, and the environment is reset when all lives are lost. Upon losing a life, a reward of $-1$ is issued as a penalty. Furthermore, in existing algorithmic solutions to episodic Atari testbeds, the rewards are transformed into $-1, 0$, or $1$ by taking their sign for stable learning, though performance is evaluated based on the original rewards. We treat the transformed rewards as the actual rewards in our continuing testbeds, removing such inconsistency.

For each testbed-algorithm pair, we performed ten runs, and each run consisted of 3M steps for Mujoco testbeds and 5M steps for Atari testbeds. The learning curves corresponding to the best parameter setting for Mujoco and Atari testbeds are shown in Figure 1 (middle row) and Figure 3, respectively. The results show that SAC and DQN consistently perform the best in Mujoco testbeds and Atari testbeds, respectively.

| | Task | DDPG | | TD3 | | SAC | | PPO | |
|---|---|---|---|---|---|---|---|---|---|
| | | episodic | continuing | episodic | continuing | episodic | continuing | episodic | continuing |
| Reward rate | HalfCheetah | **13.48 ± 0.15** | 12.19 ± 1.41 | 9.72 ± 0.57 | **10.48 ± 1.69** | 11.64 ± 1.67 | **14.23 ± 0.77** | 3.57 ± 0.57 | **3.04 ± 0.74** |
| | Ant | -0.85 ± 0.30 | **6.79 ± 0.37** | 4.74 ± 0.26 | **6.78 ± 0.09** | 5.13 ± 0.82 | **7.58 ± 0.20** | **4.48 ± 0.29** | 3.61 ± 0.47 |
| | Hopper | 3.60 ± 0.05 | **4.05 ± 0.06** | 3.77 ± 0.05 | **4.07 ± 0.04** | 3.93 ± 0.05 | **4.19 ± 0.07** | 3.83 ± 0.07 | **4.02 ± 0.07** |
| | Humanoid | 5.55 ± 0.19 | **6.50 ± 0.60** | 5.83 ± 0.11 | **7.75 ± 0.44** | 6.34 ± 0.07 | **8.09 ± 0.09** | 5.25 ± 0.03 | **7.65 ± 0.08** |
| | Walker2d | 3.72 ± 0.17 | **4.88 ± 0.19** | **4.82 ± 0.20** | 4.37 ± 0.50 | 3.05 ± 0.88 | **4.06 ± 0.83** | **5.23 ± 0.22** | 4.87 ± 0.29 |
| Number of resets | HalfCheetah | **0.50 ± 0.31** | 1.80 ± 0.96 | **0.30 ± 0.30** | 7.50 ± 5.67 | **1.20 ± 0.44** | 0.70 ± 0.30 | **0.20 ± 0.13** | 0.40 ± 0.31 |
| | Ant | 18.90 ± 8.49 | 23.00 ± 2.67 | 2.50 ± 0.87 | **1.20 ± 0.29** | **2.60 ± 0.99** | 5.70 ± 2.95 | 5.80 ± 1.16 | **4.50 ± 1.52** |
| | Hopper | **27.20 ± 1.68** | 45.50 ± 1.92 | **3.40 ± 1.90** | 45.90 ± 1.60 | **11.10 ± 2.25** | 46.90 ± 2.42 | **16.90 ± 2.52** | 52.90 ± 1.88 |
| | Humanoid | 80.70 ± 59.83 | 228.10 ± 75.23 | **0.10 ± 0.10** | 55.30 ± 20.32 | **1.00 ± 0.42** | 5.50 ± 1.93 | 61.70 ± 3.94 | 107.40 ± 3.96 |
| | Walker2d | **30.80 ± 2.44** | 42.50 ± 11.94 | **3.30 ± 1.04** | 35.30 ± 15.76 | 89.30 ± 32.12 | 103.70 ± 69.34 | **5.20 ± 0.70** | 28.70 ± 6.15 |

Table 2: A comparison of the policy learned in the continuing task vs the policy learned in the corresponding episodic task. The upper group shows the mean and the standard error of the reward rates when deploying the learned policies obtained in these two settings for $10,000$ steps. The higher reward rate is marked in boldface, and the number obtained in other settings is also marked in bold if the difference is statistically insignificant. The lower group shows the number of resets within the evaluation steps. The reset number for the fewer is marked in boldface. This table shows that policies learned in continuing tasks make more frequent resets and achieve a higher reward rate.

As mentioned earlier, when resets are predefined, the agent may choose to solve a continuing or episodic task. We now illustrate the difference between these two choices by showing the difference between policies learned in these two tasks. The episodic tasks are the same as the above continuing tasks, except that the agent optimizes cumulative rewards only up to resetting. Table 2 shows the final reward rate and the number of resets when running in the continuing tasks for $10,000$ steps, the policies learned in the continuing and episodic Mujoco tasks. The results for Atari tasks demonstrate a similar trend as in Mujoco tasks and are shown in Table 13 (Appendix B).

Table 2 demonstrates that in most cases, learned policies in continuing tasks result in higher reward rates and more resets. This likely occurs because the reset cost is relatively small compared to the additional rewards gained through aggressive actions, which have a higher likelihood of causing resets. A follow-up experiment revealed that when a large reset cost is used, fewer resets are observed in most cases, and the reward rate, surprisingly, remains comparable in most instances and even higher in some, as shown in Table 3. This suggests that reset cost functions not only as a problem parameter but also as a solution parameter that requires tuning when applying current algorithms. Future research is needed to understand how to select this solution parameter.

| | Task | DDPG | | TD3 | | SAC | | PPO | |
|---|---|---|---|---|---|---|---|---|---|
| | Reset cost | 1 | 100 | 1 | 100 | 1 | 100 | 1 | 100 |
| Reward rate (excluding reset cost) | HalfCheetah | $11.30 \pm 1.35$ | $10.46 \pm 0.27$ | $8.04 \pm 1.79$ | $6.15 \pm 1.22$ | $15.26 \pm 0.30$ | $13.28 \pm 1.47$ | $3.95 \pm 0.39$ | $3.83 \pm 0.44$ |
| | Ant | $4.26 \pm 0.07$ | $3.34 \pm 0.16$ | $2.02 \pm 0.23$ | $2.39 \pm 0.23$ | $7.26 \pm 0.14$ | $6.30 \pm 0.60$ | $2.82 \pm 0.44$ | $4.94 \pm 0.15$ |
| | Hopper | $2.85 \pm 0.03$ | $2.86 \pm 0.04$ | $2.75 \pm 0.05$ | $2.88 \pm 0.04$ | $3.93 \pm 0.13$ | $4.30 \pm 0.04$ | $3.96 \pm 0.08$ | $4.06 \pm 0.08$ |
| | Humanoid | $6.88 \pm 0.31$ | $8.02 \pm 0.37$ | $6.96 \pm 0.45$ | $8.02 \pm 0.19$ | $7.91 \pm 0.19$ | $7.51 \pm 0.27$ | $7.63 \pm 0.08$ | $6.12 \pm 0.06$ |
| | Walker2d | $3.79 \pm 0.14$ | $3.95 \pm 0.11$ | $2.64 \pm 0.38$ | $2.80 \pm 0.46$ | $4.70 \pm 0.87$ | $5.79 \pm 0.19$ | $5.10 \pm 0.22$ | $5.23 \pm 0.18$ |
| Number of resets | HalfCheetah | $2.20 \pm 1.48$ | $1.00 \pm 0.33$ | $8.10 \pm 5.10$ | $2.80 \pm 1.91$ | $0.20 \pm 0.13$ | $0.40 \pm 0.16$ | $36.30 \pm 29.99$ | $1.30 \pm 0.47$ |
| | Ant | $94.20 \pm 5.98$ | $65.20 \pm 4.76$ | $89.80 \pm 10.27$ | $58.40 \pm 9.65$ | $2.80 \pm 1.17$ | $4.80 \pm 4.37$ | $80.50 \pm 28.45$ | $4.80 \pm 1.10$ |
| | Hopper | $84.30 \pm 1.83$ | $69.70 \pm 1.74$ | $100.20 \pm 4.98$ | $86.60 \pm 3.82$ | $57.30 \pm 5.58$ | $35.80 \pm 1.14$ | $53.40 \pm 1.54$ | $44.00 \pm 2.01$ |
| | Humanoid | $161.90 \pm 52.15$ | $76.40 \pm 46.33$ | $138.00 \pm 38.23$ | $3.40 \pm 1.82$ | $44.00 \pm 15.10$ | $2.67 \pm 1.50$ | $118.30 \pm 4.72$ | $83.10 \pm 3.52$ |
| | Walker2d | $104.50 \pm 17.33$ | $39.70 \pm 3.98$ | $108.90 \pm 24.23$ | $55.40 \pm 11.98$ | $99.20 \pm 71.06$ | $3.00 \pm 1.14$ | $27.70 \pm 7.31$ | $10.70 \pm 1.24$ |

Table 3: The table presents the reward rate and number of resets of the learned policies over 10,000 evaluation steps with varying reset costs. To ensure a fair comparison, the reset cost is excluded from the reward rate computation. The lower section of the table shows the number of resets during evaluation. The boldface represents the same meaning as in Table 2. These results demonstrate that policies learned in tasks with higher reset costs generally lead to fewer resets. In several cases (e.g., DDPG in Humanoid), higher reset costs are also associated with higher reward rates.

## 2.3 TESTBEDS WHERE THE AGENT CONTROLS RESETS

This section studies the behavior of current algorithms in continuing tasks where predefined resets are not available, and the agent decides when to reset. Intuitively, allowing the agent to choose when to reset can lead to higher reward rates compared to predefined resets, as the agent can optimize its behavior by avoiding unnecessary resets. However, predefined resets reduce the state and action spaces, making the testbeds easier. For instance, in environments like Humanoid, Walker, and Hopper, the agent needs to carefully control its actions to avoid falling, and recovering from these fallen states is difficult or impossible. In such cases, the agent must learn to recognize when it cannot recover and needs to reset the environment to continue. Predefined resets simplify the problem by eliminating these bad, unrecoverable states, allowing the agent to focus on learning in good states.

The testbeds are the five Mujoco testbeds used in Section 2.2 without predefined resets. In these new testbeds, the agent can choose to reset the environment at any time step. This is achieved by augmenting the environment's action space in these testbeds by adding one more dimension. This additional dimension has a range of $[0, 1]$, representing the probability of reset. The tested continuous control algorithms can then be readily applied, except that the exploration noise for this additional dimension needs to be set differently from other action dimensions because the performance of the policy is more sensitive to this dimension than the rest. We leave the details of the tested noises in Section A.3. The number of runs and number of steps in each run are chosen in the same way as in the above two subsections. The tested hyperparameters are provided in Section A.2. The learning curves, which are chosen the same way as the previous two subsections, are reported in Figure 1 (lower row) (Appendix B). We also show in Table 14 (Appendix B) the reward rate and the number of resets achieved by the final learned policy deployed for $10,000$ steps and compare it to the reward rates when the policies are learned in the testbeds with predefined resets.

Comparing the performance of the tested algorithms in testbeds with predefined resets and those with agent-controlled resets reveals some nuanced results. In many cases, algorithms trained in testbeds with agent-controlled resets achieved a similar final reward rate to those with predefined resets. In a few instances, algorithms in testbeds with agent-controlled resets performed better, achieving both higher final reward rates and more stable learning (e.g., PPO in HalfCheetah and Ant). Conversely, in other cases, the learned policies performed worse. Notably, some learning curves show a significant upward trend toward the end of training, suggesting that the performance differences may be due, at least in part, to the larger state and action spaces in the testbeds with agent-controlled resets, which could require more training time to fully optimize. Nevertheless, longer training time does not always suffice. For instance, in the Humanoid task, all algorithms performed considerably worse when resets were learned. The learning curves for most algorithms, except PPO, demonstrate slow improvement over time. DDPG faced such challenges that its final learned policy was even worse than the performance of a random policy in the Humanoid task with predefined resets (approximately 4.6). The failure in Humanoid likely stems from the fact that it has a significantly larger state space compared to other testbeds.

### 2.4 FAILURE TO ADDRESS LARGE DISCOUNT FACTORS OR OFFSETS IN REWARDS

Using the Mujoco testbeds presented above, we show in this section that the performance of all of the tested continuous control algorithms deteriorates significantly when a large discount factor is used or when all rewards are shifted by the large constant.

We report the percentage of improvement for each testbed-algorithm pair, defined as $\frac{\bar{r}^{0.999} - \bar{r}^{\mathrm{random}}}{\bar{r}^{0.99} - \bar{r}^{\mathrm{random}}} - 1$, where $\bar{r}^{0.999}$ is the final average reward rate over the last

|  |  | Discount factor 0.99 → 0.999 | | | | All rewards +100 | | | |
|---|---|---|---|---|---|---|---|---|---|
|  | Algorithm | DDPG | TD3 | SAC | PPO | DDPG | TD3 | SAC | PPO |
| No resets | Swimmer | -85.95 | -45.19 | -99.23 | 46.84 | -104.86 | -103.20 | -108.30 | -101.52 |
|  | HumanoidStandup | -9.09 | 14.16 | -60.13 | -13.45 | -29.16 | 5.70 | -24.10 | -11.97 |
|  | Reacher | -707.42 | -6.01 | -10.13 | 1.60 | -429.94 | -160.87 | -117.87 | -8.67 |
|  | Pusher | -13.80 | -10.82 | -7.23 | -4.54 | -183.44 | -162.26 | -25.07 | -19.53 |
|  | SpecialAnt | -38.39 | -67.71 | -152.16 | -11.86 | -100.50 | -44.65 | -12.73 | -42.30 |
| Predefined resets | HalfCheetah | -20.62 | 49.84 | 4.26 | -41.32 | -59.69 | -85.34 | -44.86 | -62.95 |
|  | Ant | -7.48 | -22.46 | -14.66 | -15.50 | -118.93 | -97.01 | -75.70 | -31.90 |
|  | Hopper | -12.81 | -8.45 | -11.72 | -21.73 | -62.35 | -53.05 | -17.77 | -36.00 |
|  | Humanoid | -34.28 | -64.27 | -74.81 | -58.51 | -83.17 | -113.79 | -109.34 | -50.57 |
|  | Walker2d | -5.07 | -15.12 | -3.38 | -29.89 | -63.41 | -53.33 | -40.23 | -52.50 |
| Agent-controlled resets | HalfCheetah | -27.19 | -29.79 | -33.93 | -26.41 | -73.96 | -26.31 | -59.21 | -78.05 |
|  | Ant | -5.32 | 10.74 | -22.89 | -19.04 | -127.31 | -85.00 | -82.81 | -69.68 |
|  | Hopper | -29.62 | -11.62 | -5.92 | -12.87 | -106.48 | -65.56 | -11.30 | -36.35 |
|  | Humanoid | -155.59 | 2.13 | -14.77 | -23.05 | -106.27 | -108.54 | -35.26 | -21.05 |
|  | Walker2d | -30.49 | 6.14 | -37.61 | -22.96 | -59.64 | -46.72 | -35.39 | -77.86 |

Table 4: A large discount factor or reward offset hurt all tested algorithms' performance.

10,000 steps with a discount factor of 0.999, and $\bar{r}^{0.99}$ is the reward rate with a discount factor of 0.99. The term $\bar{r}^{\mathrm{random}}$ refers to the reward rate of a uniformly random policy. As in the previous subsections, all reward rates are averaged over ten runs, each of which has 3 million steps, and gray shading indicates that the difference between $\bar{r}^{0.99}$ and $\bar{r}^{0.999}$ is not statistically significant, as determined by Welch's t-test with a $p$-value less than 0.05. Additionally, we tested these pairs when all environment rewards were shifted by +/-100, with other experiment details the same as above. We report the percentage of improvement computed in a similar way as for discount factors when all environment rewards are shifted by +100 but with the common offset subtracted for a fair comparison. Formally, this percentage of improvement is $\frac{\bar{r}^{100} - 100 - \bar{r}^{\mathrm{random}}}{\bar{r} - \bar{r}^{\mathrm{random}}} - 1$, where $\bar{r}^{100}$ is the final average reward rate over the last 10,000 steps when all rewards are shifted by +100, and $\bar{r}$ is the reward rate without reward shifting. The results when all rewards are subtracted by -100 are similar and are thus omitted. The results (Table 4) show that, overall, algorithms with a discount factor of 0.999 perform much worse than those with 0.99. Moreover, a large reward offset leads to catastrophic failure across almost all task-algorithm pairs.

## 3 EVALUATING ALGORITHMS WITH REWARD CENTERING

This section empirically shows that the temporal-difference-based reward centering method, originally introduced by Naik et al. (2024), improves or maintains the performance of all tested algorithms in the testbeds introduced in the previous section. Further, this method mitigates the negative effect when using a large discount factor and completely removes the detrimental effect caused by a large common reward offset.

The idea of reward centering stems from the following observation. By Laurent series expansion (Puterman, 2014), if a policy $\pi$ results in a Markov chain with a single recurrent class, its discounted value function $v_\pi$ can be decomposed into two parts, a state-independent offset $d_\pi^\top v_\pi = r(\pi)/(1-\gamma)$, where $d_\pi$ is the stationary distribution under $\pi$, $r(\pi)$ is the average reward rate under policy $\pi$, and a state-dependent part keeping the relative differences among states. Here, the reward rate does not depend on the initial state due to the assumption of a single recurrent class. Note that only the state-dependent part is useful for improving the policy $\pi$. However, when the state-independent part has a large magnitude, possibly due to large offsets in rewards or a discount factor that is close to 1, approximating the state-independent part separately for each state can result in approximation errors that mask the useful state-dependent part.

Reward centering approximates the state-independent part using a shared scalar. Specifically, reward centering approximates a new discounted value function, obtained by subtracting all rewards by an approximation of $r(\pi)$, and this new discounted value function has a zero state-independent offset if the approximation of $r(\pi)$ is accurate. Even if the approximation of $r(\pi)$ is not accurate, removing a portion of the state-independent offset still helps.

| | Task | DDPG | TD3 | SAC | PPO |
|---|---|---|---|---|---|
| No resets | Swimmer | 109.11 | 90.71 | 1149.26 | 71.14 |
| | HumanoidStandup | 41.67 | 19.79 | 35.83 | 19.39 |
| | Reacher | -0.03 | 0.07 | -0.11 | 1.17 |
| | Pusher | 10.87 | 1.24 | 0.39 | 3.72 |
| | SpecialAnt | 12.67 | 2.05 | 5.59 | 10.55 |
| Predefined resets | HalfCheetah | 3.15 | 13.13 | 5.05 | 4.66 |
| | Ant | 22.22 | 18.25 | 6.75 | 13.36 |
| | Hopper | 2.53 | 14.83 | 4.44 | 4.56 |
| | Humanoid | 210.54 | 89.68 | 77.54 | 11.29 |
| | Walker2d | 16.28 | 10.37 | 7.72 | 7.37 |
| Agent-controlled resets | HalfCheetah | 2.21 | 17.45 | -2.05 | 4.06 |
| | Ant | 12.73 | 94.13 | 34.57 | 8.07 |
| | Hopper | 42.28 | 15.72 | 4.60 | 5.57 |
| | Humanoid | 246.73 | 20.71 | 5.46 | 2.36 |
| | Walker2d | 10.23 | 12.92 | 0.89 | 4.61 |
| | Average improvement | 49.54 | 28.07 | 89.06 | 11.46 |

Table 5: Percentage of reward rate improvement when applying reward centering to the tested algorithms in Mujoco testbeds.

A straightforward way to perform reward centering is to estimate $r(\pi)$ using an exponential moving average of all observed rewards. For on-policy algorithms, this moving average approach can guarantee convergence to $r(\pi)$. However, for off-policy algorithms, this approach does not converge to $r(\pi)$ (e.g., the behavior policy is uniformly random while the target policy is deterministic).

We now briefly describe the TD-based reward-centering approach, which can be applied to both on- and off-policy algorithms. This approach extends an approach to solve the average-reward criterion (Wan et al., 2021) to the discounted setting. Here, we illustrate this approach using use TD(0) (Sutton, 2018, p. 120), the simplest TD algorithm, as an example. More details on how tested algorithms employ this approach are provided in Section A.4.

Given transitions $(S, R, S')$ generated by following some policy $\pi$, TD(0) estimates $v_\pi$ by maintaining a table of value estimates $V : \mathbb{R}^{|S|}$ and updating them using $V(S) \leftarrow V(S) + \alpha\delta$, where $\delta \overset{\text{def}}{=} R + \gamma V(S') - V(S)$ is a TD error, $\alpha$ is a step-size parameter, and $\gamma$ is a discount factor. The TD-based reward-centering approach simply replaces the above TD error in TD(0) with the following new TD error:

$$\delta^{\text{RC}} \overset{\text{def}}{=} R - \bar{R} + \gamma V(S') - V(S),$$

where $\bar{R}$, a biased estimate of the reward rate, is also updated by the TD error $\delta^{\text{RC}}$, as follows:

$$\bar{R} \leftarrow \bar{R} + \eta\alpha\delta^{\text{RC}},$$

where $\eta > 0$ is a constant. It is straightforward to show that, under certain asynchronous stochastic approximation assumptions on $\alpha$, $V(s)$ converges to $v_\pi(s) - \frac{\eta}{1-\gamma+\eta|S|} \sum_{s\in S} v_\pi(s)$, following the same steps as in the proof of Theorem 1 by Naik et al. (2024). This result implies that although TD-based reward centering does not fully remove the state-independent offset $d_\pi^\top v_\pi$, it can remove a significant portion of it. Empirically, we also observed this effect.

| Task | DQN | SAC | PPO |
|---|---|---|---|
| Breakout | -7.48 | 1.67 | 11.51 |
| Pong | 0.50 | 51.94 | 79.18 |
| SpaceInvader | 20.97 | 0.29 | 19.72 |
| BeamRider | 7.01 | 35.00 | 75.67 |
| Seaquest | 26.79 | 22.75 | 5.77 |
| MsPacman | 9.96 | 1.76 | 2.67 |
| Average improvement | 9.63 | 18.90 | 32.42 |

Table 6: Percentage of reward rate improvement when applying reward centering to the tested algorithms in Atari tasks. Statistically significant improvement percentage numbers are marked in boldface.

We evaluated algorithms with TD-based reward centering across all testbeds, comparing them to base algorithms that do not use reward centering. Each experiment was repeated ten times with different seeds, lasting 1 million steps for Mujoco testbeds and 5 million steps for Atari testbeds. We report the percentage improvement when using reward centering. Specifically, the reported number is $\frac{\bar{r}^{\text{RC}} - \bar{r}^{\text{random}}}{\bar{r} - \bar{r}^{\text{random}}} - 1$, where $\bar{r}^{\text{RC}}$ is the average of all received rewards, averaged across ten runs, with TD-based reward centering, $\bar{r}$ is defined similarly but without reward centering, and $\bar{r}^{\text{random}}$ is average-reward rate of a uniformly random policy. The reported value is the best result across all tested hyperparameter settings for both reward-centered and baseline algorithms. Shaded values indicate that performance differences are not statistically significant, according to Welch's t-test with $p < 0.05$. The reported results for Mujoco and Atari testbeds are shown in Table 5 and Table 6, respectively. The corresponding learning curves are provided in Appendix C. These results show that reward centering improves or maintains the performance of all of the tested algorithms in all testbeds. How much the performance improvement seems to depend on both the algorithm and the task.

In Tables 15 and 16, we show, using the Mujoco testbeds, that TD-based reward centering is most effective when using a large discount factor or when there is a large offset in rewards, echoing the findings by Naik et al. (2024) about DQN in smaller scale testbeds. However, unlike Naik et al.'s (2024) results, our results show that while the negative effect of large discount factors is much smaller with reward centering, it can still lead to notably worse performance in many cases. This suggests that when applying tested algorithms to solve complex continuing tasks, tuning the discount factor may still be valuable, even with reward centering.

We also evaluated the exponential moving average approach to perform reward centering. In addition, we evaluated another reward-centering approach inspired by Devraj and Meyn's (2021) relative Q-learning algorithms. The details of these two approaches are provided in Section A.4. The results in Tables 17 and 18 show that the moving-average-based approach works surprisingly well despite its theoretical unsoundness in off-policy algorithms, the reference-state-based approach helps in some cases while hurts the performance in some others, and the TD-based approach is the more effective than the other two.

## 4 CONCLUSIONS AND LIMITATIONS

This paper empirically examines the challenges that continuing tasks with various reset scenarios pose to several well-known deep RL algorithms, using a suite of testbeds based on Mujoco and Atari environments. Our findings highlight key issues that future algorithmic advancements for continuing tasks may focus on. For instance, we demonstrate that the performance of tested algorithms can heavily depend on the availability of predefined resets, as these resets help agents escape traps and reduce the state space complexity. When predefined resets are available, all algorithms perform reasonably well, learning policies that exploit frequent resetting to achieve higher rewards. The reset cost balances this trade-off and also functions as a tuning parameter. In contrast, agent-controlled reset tasks are generally more challenging, and in some testbeds, allowing the agent to control resets significantly worsens performance. Additionally, we show that both a large discount factor and a large common offset in rewards can negatively impact the performance of all tested algorithms. Our results also validate the effectiveness of an existing approach to address these issues, demonstrating through extensive experiments that the negative impact of reward offset can be completely eliminated, while the harm from a large discount factor can be largely mitigated with a TD-based reward-centering approach. Even in scenarios with a smaller discount factor and no reward offset, this approach shows benefits across many testbeds for all tested algorithms.

This paper has several limitations. First, this paper focuses exclusively on the performance of online RL algorithms, leaving research on offline RL algorithms in continuing tasks unexplored. Second, although we concentrate on well-known discounted algorithms, it is worth investigating whether average-reward algorithms, such as those mentioned in Section 1, face similar challenges. Third, while most of the hyperparameters used in the experiments are standard choices and have been effective in episodic testbeds, they may not be ideal for continuing tasks. Identifying hyperparameter choices that are more suitable for continuing tasks remains unexplored. Despite these limitations, we believe that our findings provide valuable insights into the challenges of continuing tasks in deep RL, and they serve as a basis for future research.

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
