# A DETAILS OF EXPERIMENT SETUP

This appendix provides details on the experiments conducted to produce the results presented in the main text and the subsequent two appendices. First, we present additional hyperparameters used by the algorithms tested in testbeds without resets or with predefined resets. Next, we describe the modifications made to the tested algorithms for testbeds with agent-controlled resets. Following this, we provide detailed information about how the tested algorithms are used together with reward centering. In the main text, we introduced the TD-based reward centering approach; here, we describe two additional approaches for performing reward centering, outlining how these methods were applied to the tested algorithms, along with the values of additional hyperparameters tested for reward centering.

## A.1 AVERAGE-REWARD RATE AS THE EVALUATION METRIC

In reinforcement learning (RL), an agent interacts with an environment to learn how to make decisions that maximize a cumulative reward signal. The environment is typically modeled as a finite Markov Decision Process (MDP), which consists of a tuple $(\mathcal{S}, \mathcal{A}, \mathcal{R}, p)$, where $\mathcal{S}$ represents the set of states, $\mathcal{A}$ the set of actions, $\mathcal{R}$ is the set of rewards, $p(s', r \mid s, a)$ is the probability of transitioning from state $s$ to $s'$ and observing a reward of $r$, given action $a$. At each time step $t$, the agent observes the current state $S_t$, selects an action $A_t$ based on a policy, and receives a reward signal $R_{t+1}$ from the environment, with the goal of learning a policy that maximizes long-term reward.

For continuing tasks, where the agent-environment interaction persists indefinitely, the average-reward criterion is suitable as the performance metric and is therefore used in this paper. Let the initial state be $s_0$, the average reward is defined as $r(\pi, s_0) \overset{\text{def}}{=} \lim_{T \to \infty} \frac{1}{T} \mathbb{E}\left[\sum_{t=1}^{T} R_t \Big| A_t \sim \pi(\cdot \mid S_t), S_0 = s_0\right]$, where $\pi : \mathcal{S} \to \Delta(\mathcal{A})$ is the agent's policy.

While there are several deep RL algorithms (e.g., Zhang and Ross 2021) addressing the average-reward criterion, we choose to study several well-known discounted deep RL algorithms. This is because the focus of this paper is on the challenges of continuing tasks rather than on studying the properties of algorithms, and these discounted algorithms have been better understood in the literature. Further, note that by adjusting the discount factor to be close to one, discounted algorithms can approximately solve the average-reward criterion in continuing tasks. When the discount factor is sufficiently close to one, any discounted optimal policy is also average-reward optimal (Grand-Clément and Petrik, 2024).

## A.2 TESTED HYPERPARAMETER FOR ALGORITHMS IN TESTBEDS WITHOUT RESETS OR WITH PREDEFINED RESETS

We provide hyperparameters used by the tested algorithms in Tables 7—12.

| Hyperparameter | Value |
| --- | --- |
| Actor & critic networks | fully connected with $256 \times 256$ hidden layers and Relu activation |
| Optimizer | Adam |
| Discount factor | 0.99, 0.999 |
| Actor & critic learning rates | 3e-4 |
| Actor & critic target smoothing coefficients | 0.005 |
| Batch size | 256 |
| Replay buffer size | 1e6 |
| Exploration noise distribution | Normal(0, 0.1) |
| Warmup stage (taking random actions) | first 25000 steps |
| Learning after | first 25000 steps |

Table 7: Tested DDPG and TD3's hyperparameters for Mujoco testbeds. TD3, in addition, makes a delayed actor update every other critic update. The noise added in the sample action used in TD3's update is a zero mean Normal distribution with noise $0.2$. This noised sampled action is then clipped to be within $[-0.5, 0.5]$.

| Hyperparameter | Value |
| --- | --- |
| Actor & critic networks | fully connected with $256 \times 256$ hidden layers and Relu activation |
| Optimizer | Adam |
| Discount factor | 0.99, 0.999 |
| Actor learning rate | 3e-4 |
| Critic learning rate | 1e-3 |
| Use autotune | True |
| Critic target smoothing coefficient | 0.005 |
| Batch size | 256 |
| Replay buffer size | 1e6 |
| Warmup stage (taking random actions) | first 5000 steps |
| Learning after | first 5000 steps |

Table 8: Tested SAC's hyperparameters for Mujoco testbeds

| Hyperparameter | Value |
| --- | --- |
| Actor & critic networks | fully connected with $64 \times 64$ hidden layers and Tanh activation |
| Optimizer | Adam |
| Discount factor | 0.99, 0.999 |
| Actor & critic learning rates | 3e-4 |
| $\lambda$ in generalized advantage estimation | 0.95 |
| Importance sampling ratio clipping range | [0.8, 1.2] |
| Samples collected for updates | 2048 |
| Batch size | 64 |
| Number of updates per sample | 10 |
| Gradient norm clipping threshold | 0.5 |
| Normalize advantage | True |
| Value clipping | False |
| Return normalization | False |
| Entropy coefficient | 0.0 |

Table 9: Tested PPO's hyperparameters for Mujoco testbeds

| Hyperparameter | Value |
|---|---|
| Q network | three convolution layers followed by one fully connected layer, all with Relu activation |
| Conv layer 1 | kernel size 8, output channel size 32, strides 4, paddings 0 |
| Conv layer 2 | kernel size 4, output channel size 64, strides 2, paddings 0 |
| Conv layer 3 | kernel size 3, output channel size 64, strides 1, paddings 0 |
| Fully connected layer size | 512 |
| Optimizer | Adam |
| Discount factor | 0.99, 0.999 |
| Learning rate | 1e-4 |
| Replay buffer size | 800000 |
| Samples between two updates | 4 |
| Target network update | every 1000 steps |
| Batch size | 64 |
| Number of updates per sample | 10 |
| Normalize advantage | True |
| Gradient norm clipping threshold | 0.5 |
| Exploration | $\epsilon$-greedy with linear decay. $\epsilon$ starts from $\epsilon = 1$ and ends at 0.01. 1000000 decay steps. |
| Learning after | first 80000 steps |

Table 10: Tested hyperparameters for DQN for Atari testbeds.

| Hyperparameter | Value |
|---|---|
| Actor & critic networks | three convolution layers followed by one fully connected layer, all with Relu activation |
| Conv layer 1 | kernel size 8, output channel size 32, strides 4, paddings 0 |
| Conv layer 2 | kernel size 4, output channel size 64, strides 2, paddings 0 |
| Conv layer 3 | kernel size 3, output channel size 64, strides 1, paddings 0 |
| Fully connected layer size | 512 |
| Optimizer | Adam |
| Discount factor | 0.99, 0.999 |
| Actor & critic learning rate | 3e-4 |
| $\lambda$ in generalized advantage estimation | 0.95 |
| Importance sampling ratio clipping range | [0.9, 1.1] |
| Samples collected for updates | 1024 |
| Batch size | 256 |
| Number of updates per sample | 8 |
| Gradient norm clipping threshold | 0.5 |
| Normalize advantage | True |
| Value clipping | False |
| Return normalization | False |
| Entropy coefficient | 0.01 |

Table 11: Tested hyperparameters for PPO for Atari testbeds.

| Hyperparameter | Value |
|---|---|
| Actor & critic networks | three convolution layers followed by one fully connected layer, all with Relu activation |
| Conv layer 1 | kernel size 8, output channel size 32, strides: 4, paddings: 0 |
| Conv layer 2 | kernel size 4, output channel size 64, strides: 2, paddings: 0 |
| Conv layer 3 | kernel size 3, output channel size 64, strides: 1, paddings: 0 |
| Fully connected layer size | 512 |
| Optimizer | Adam |
| Discount factor | 0.99, 0.999 |
| Actor & critic learning rates | 3e-4 |
| Use autotune | False |
| Entropy coefficient | 0.2 |
| Samples collected between two updates | 4 |
| Critic target update frequency | 2000 |
| Batch size | 64 |
| Replay buffer size | 800000 |
| Warmup stage (taking random actions) | first 20000 steps |
| Learning after | first 20000 steps |

Table 12: Tested hyperparameters for SAC for Atari testbeds.

### A.3 HYPERPARAMETERS WHEN APPLIED TO TESTBEDS WITH AGENT-CONTROLLED RESETS

We modified the hyperparameters of the tested algorithms in two ways to improve the algorithms' performance in testbeds with agent-controlled resets.

First, we adjust a hyperparameter that controls the level of exploration for DDPG, TD3, and SAC. For DDPG and TD3, the exploration noise is a sample of a zero-mean multivariate Gaussian random vector with independent elements. This exploration noise is then added to the action generated by the actor network to perform persistent exploration. For testbeds without resets or with predefined resets, we applied the same standard deviation of 0.1 to all elements. However, when resets are part of actions, we tested smaller standard deviations, including 0.05, 0.005, 0.0005, and 0.00005, for the reset dimension. This is because, compared to the other dimensions in actions, a small noise in the reset dimension would have a significant effect on the behavior of the policy. For SAC, the entropy regularization coefficient controls the level of exploration. We applied the autotune technique introduced by Haarnoja et al. (2018) to adjust this coefficient dynamically. This technique introduces some regularization that pushes the entropy of the learned policy toward some predefined target value, guaranteeing that exploration does not diminish to zero asymptotically. For testbeds without resets or with predefined resets, the target entropy was chosen to be $-|\mathcal{A}|$, a choice tested by Haarnoja et al. (2018), where $|\mathcal{A}|$ is the dimension of the action space. When resetting is part of the action, we found this choice leads to very frequent resets, even at the end of training. We therefore tested smaller target entropy values, including $-|\mathcal{A}|, -|\mathcal{A}| - 3, -|\mathcal{A}| - 6$, and $-|\mathcal{A}| - 9$. PPO's exploration noise is learned, and there is no mechanism for maintaining exploration above a certain level or pushing exploration toward a certain level. Therefore, no more changes need to be applied to PPO's hyperparameters.

The second change we made was to have a different random policy for collecting data in the warmup stage of DDPG, TD3, and SAC. In testbeds without resets or with predefined resets, a policy that uniformly randomly samples from the action space was used in the warmup stage. When resetting probability is part of the action, we apply a different policy that is biased toward lower reset probability. The reason is that a uniformly random policy would output a reset probability of 0.5, which is so high that most of the data collected following this policy will be several steps away from the initial states. To generate longer trajectories, we chose the resetting probability element of the action to be $1/N$, where $N$ is an integer sampled uniformly from $1, 2, \ldots, 1000$, and kept other elements uniformly sampled.

### A.4 APPLYING REWARD CENTERING METHODS TO THE TESTED ALGORITHMS

In this section, we describe how we applied three reward-centering approaches to the tested algorithms. We start with TD-based reward centering and then discuss two other alternative approaches.

We apply TD-based reward centering to DQN the same way as Naik et al. (2024) did. DQN maintains an approximate action-value function, $q_w : \mathcal{S} \times \mathcal{A} \to \mathbb{R}$, with the vector $w$ being the parameters of the function. To update $w$, DQN maintains a target network $q_{\hat{w}}$ parameterized the same way as $q_w$ but with different parameters. For every fixed number of time steps, the values of $w$ are copied to $\hat{w}$. Every time step, DQN samples a batch of transition tuples $(s_i, a_i, r_i, s'_i), i \in \{1, 2, \ldots, n\}$ from the replay buffer, where $s_i, a_i, r_i, s'_i$ denote a state, an action, and the resulting reward and state, respectively, and $n$ is the batch size. The update rule to $w$ is

$$w \overset{\text{def}}{=} w + \alpha \frac{1}{n} \sum_{i=1}^{n} \delta_i \nabla_w q_w(s_i, a_i), \tag{1}$$

where $\delta_i \overset{\text{def}}{=} r_i + \gamma \max_{a \in \mathcal{A}} q_{\hat{w}}(s'_i, a) - q_w(s_i, a_i)$ is a TD error, $\alpha$ is a step-size parameter and $\gamma$ is a discount factor. With TD-based reward centering, we update $w$ with equation 1 but with $\delta_i$ replaced by a different TD error, where the reward is subtracted by an offset $\bar{r}$, defined as follows:

$$\delta_i^{\text{RC}} \overset{\text{def}}{=} \delta_i - \bar{r}. \tag{2}$$

The offset $\bar{r}$ is updated whenever $w$ is updated, using the new TD errors, following

$$\bar{r} \overset{\text{def}}{=} \bar{r} + \beta \frac{1}{n} \sum_{i=1}^{n} \delta_i^{\text{RC}}, \tag{3}$$

where $\beta$ is another step-size parameter. The tested $\beta$ values in our experiments are $3e-2, 1e-2, 3e-3, 1e-3, 3e-4$. The tested discount factors in our experiments are $0.99, 0.999$, and $1.0$. These hyperparameters were also used in other tested algorithms with reward centering.

DDPG, TD3, SAC, and PPO are also driven by TD-learning with various different TD errors. We show their respective TD errors below. The centered versions of DDPG, TD3, and SAC can be derived straightforwardly by replacing their respective TD errors with the new TD errors obtained, as in equation 2, and updating $\bar{r}$ whenever their critic parameters are updated, as in equation 3. PPO requires a slightly more complicated treatment in the update of $\bar{r}$, which we will discuss separately.

Like DQN, DDPG also samples a batch of transition tuples $(s_i, a_i, r_i, s'_i), i \in \{1, 2, \ldots, n\}$ from the replay buffer to update the weight vector $w$. In addition, the algorithm samples an action $a'_i$ according to the actor's policy for each $s'_i$. DDPG's TD error is $\delta_i \stackrel{\text{def}}{=} r_i + q_{\hat{w}}(s'_i, a'_i) - q_w(s_i, a_i)$.

TD3 maintains two approximate value functions that are parameterized in the same way but with different parameters. Denote them by $q_{w_1}, q_{w_2}$. To update $w_1$ and $w_2$, for each time step, just like DDPG, TD3 samples a batch of transition tuples $(s_i, a_i, r_i, s'_i), i \in \{1, 2, \ldots, n\}$ from the replay buffer and a batch of actions $a'_i$. Unlike in DDPG, an additional Gaussian noise $\epsilon_i$ is added on $a'_i$. TD3's TD error is $r_i + \gamma \left( \min_{j \in \{1,2\}} q_{w_j}(s'_i, a'_i + \epsilon_i) \right) - q_{w_j}(s_i, a_i)$.

SAC also employs two approximate value functions $q_{w_1}, q_{w_2}$. Let $(s_i, a_i, r_i, s'_i), i \in \{1, 2, \ldots, n\}$ and $a'_i$ be generated the same way as in DDPG. The continuous control version of SAC's TD error is $r_i + \gamma \left( \min_{j \in \{1,2\}} q_{w_j}(s'_i, a'_i) \right) - \kappa \log \pi(a'_i \mid s'_i) - q_{w_j}(s_i, a_i)$, where $\kappa$ is a regularization coefficient influencing the entropy of the policy and is either predefined or automatically tuned, and $\pi$ is the actor's policy. The discrete control version of SAC does not use sampled actions $a'_i$ but considers all possible actions and uses the expectation. Its TD error is $\delta_i \stackrel{\text{def}}{=} r_i + \gamma \left( \sum_{a \in \mathcal{A}} \pi(a \mid s'_i) \left( \min_{j \in \{1,2\}} q_{w_j}(s'_i, a) \right) - \kappa \log \pi(a \mid s'_i) \right) - q_{w_j}(s_i, a_i)$.

PPO does not maintain an approximate action-value function but an approximate state-value function $v_w : \mathcal{S} \to \mathbb{R}$, with $w$ being the weight vector. PPO proceeds in rounds. For each round, PPO collects a certain number of transitions following the current policy without changing any parameters and then applies multiple updates to both actor and critic parameters using the transitions. These transitions are not used in subsequent rounds. Let $S_t, A_t$ denote the state, action at time step $t$ and let $R_{t+1}$ denote the resulting reward. PPO's TD error at time step $t$ is defined as follows:

$$\delta_t \stackrel{\text{def}}{=} R_{t+1} + \gamma v_w(S_{t+1}) - v_w(S_t).$$

From this TD error, generalized advantage estimate (GAE) and truncated $\lambda$-return are computed, which are used to update actor and critic parameters. The centered TD error for PPO is defined by $\delta_t^{\text{RC}} \stackrel{\text{def}}{=} \delta_t - \bar{r}$, as in equation 2.

However, unlike the above algorithms, the update to $\bar{r}$ is performed using all transitions collected in a round instead of a batch of transitions because this does not add too much additional computation, given that the TD errors for all transitions visited in the round need to be computed anyway, to obtain the GAE and truncated $\lambda$-return.

Regarding the update to $\bar{r}$, another difference between PPO and the above algorithms is that in PPO, $\bar{r}$ is not performed every time the critic is updated, but every time the TD error is computed, to save computation. Recall that PPO's parameter updates proceed in epochs. For each epoch, all transitions collected in the current round are used for one time in both actor and critic updates. Even if there are multiple critic updates within each epoch, the TD errors are only computed once at the beginning of each epoch.

The above discussion finishes with a discussion of how we apply TD-based reward centering to the tested algorithms. We now discuss the other two reward-centering approaches.

The first approach is to simply let $\bar{r}$ be updated with an exponential moving average of the past rewards instead of being updated by equation 3. Formally, at time step $t$, $\bar{r}$ is updated with

$$\bar{r} \leftarrow \beta\bar{r} + (1 - \beta)R_t,$$

where $\beta \in [0, 1]$ is the moving average rate. The tested $\beta$ values are $0.99, 0.999, 0.9999$. As suggested by Naik et al. (2024), this approach is theoretically sound in on-policy algorithms, such as

PPO, but is problematic for off-policy algorithms, such as the rest of the tested algorithms. In our paper, we empirically test this approach for all algorithms.

The other approach uses a set of reference states and is based on the relative Q-learning family of algorithms proposed by Devraj and Meyn (2021). These algorithms are tabular discounted algorithms and can be viewed as the extension of relative-value-iteration(RVI)-based Q-learning (Abounadi et al., 2001), a family of average-reward algorithms, to the discounted setting. Here, we briefly discuss the idea of relative Q-learning. We will then mention how to perform reward centering in the tested algorithms following the same idea.

Relative Q-learning maintains a $\mathcal{S} \times \mathcal{A}$-sized table of estimates for action values and updates these estimates in a similar way as Q-learning. For each time step, a state $S_t$ is observed, and an action $A_t$ is chosen by a policy that may or may not be controlled by the agent; the algorithm then updates with the resulting transition $(S_t, A_t, R_{t+1}, S_{t+1})$ using the following update rule:

$$Q(S_t, A_t) \leftarrow Q(S_t, A_t) + \alpha \left( R_{t+1} - f(Q) + \gamma \max_{a \in \mathcal{A}} Q(S_{t+1}, a) - Q(S_t, A_t) \right), \qquad (4)$$

where $f$ is a function satisfying certain properties. Examples of such functions include $f(Q) = \frac{1}{|\mathcal{S}| \times |\mathcal{A}|} \sum_{s \in \mathcal{S}, a \in \mathcal{A}} Q(s, a)$, $f(Q) = \max_{s \in \mathcal{S}, a \in \mathcal{A}} Q(s, a)$, or $f(Q) = \min_{s \in \mathcal{S}, a \in \mathcal{A}} Q(s, a)$. Here $f(Q)$ is a common offset subtracted by all rewards, therefore serving the same role as $\bar{r}$.

We make a few observations regarding this algorithm. First, note that the $f$ function is chosen before the agent starts and is fixed through the agent's lifetime. Second, note that, unlike $\bar{r}$, $f(Q)$ does not need to be estimated separately. Third, note that the centered Q-learning algorithm introduced by Naik et al. (2024) with tabular representation can be written in equation 4 with $f(Q) \overset{\text{def}}{=} \eta \sum_{s \in \mathcal{S}, a \in \mathcal{A}} Q(s, a)$, where $\eta$ is a hyperparameter. However, the relation between the two algorithms with function approximation is unclear.

Following the above idea, we replaced $\bar{r}$ in the tested algorithms by $f(q_w) \overset{\text{def}}{=} \frac{1}{|\mathcal{I}|} \sum_{(s,a) \in \mathcal{I}} q_w(s, a)$, where $\mathcal{I}$ is a fixed set of state-action pairs or $f(q_{w_j}) \overset{\text{def}}{=} \frac{1}{2|\mathcal{I}|} \sum_{(s,a) \in \mathcal{I}} (q_{w_1}(s, a) + q_{w_1}(s, a))$ when two value functions are used. The size of $\mathcal{I}$ is the same as the batch size used in each algorithm. The pairs are sampled randomly from the replay buffer right before the first learning update. Readers may refer to Tables 7—12 for the first learning update time step.

## B  ADDITIONAL EVALUATION RESULTS OF TESTED RL ALGORITHMS

This appendix shows evaluation results of tested RL algorithms that are omitted in Section 2 of the main text.

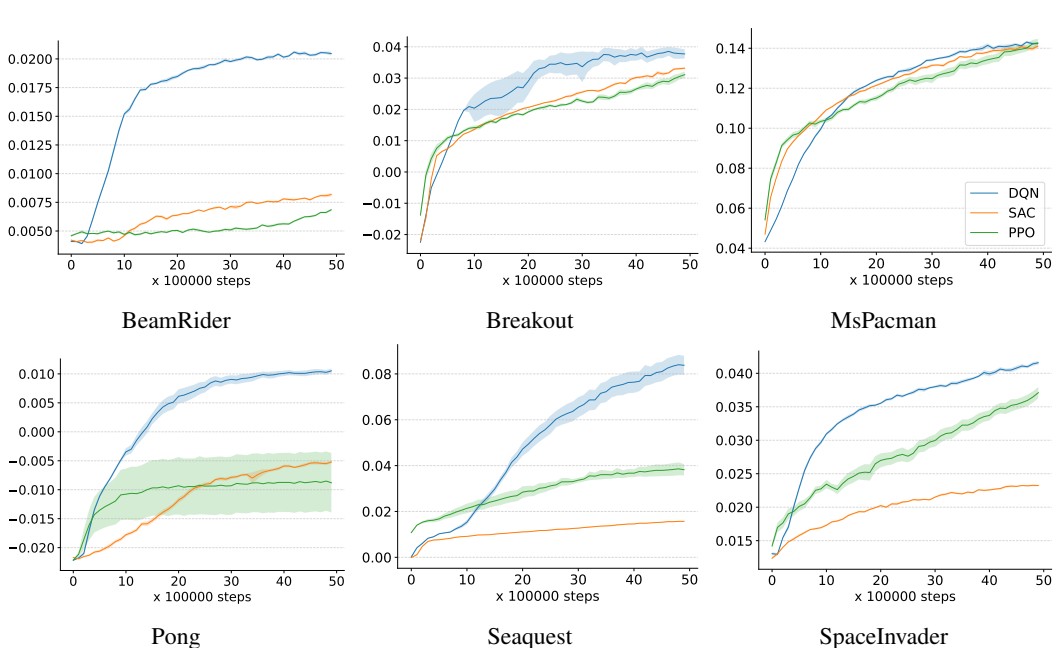

Figure 3: Learning curves in continuing testbeds with predefined resets based on the Atari environment. Each point shows the reward rate over the past 100k steps. Shading area standards for one standard error. Overall, DQN performs the best of the three tested algorithms.

|  |  | DQN | | SAC | | PPO | |
|---|---|---|---|---|---|---|---|
|  |  | episodic | continuing | episodic | continuing | episodic | continuing |
| Reward rate | Breakout | **3.43 ± 0.36** | **4.01 ± 0.21** | 2.08 ± 0.61 | **3.51 ± 0.08** | **3.02 ± 0.07** | **3.05 ± 0.13** |
|  | Pong | 0.76 ± 0.07 | **1.01 ± 0.07** | -0.64 ± 0.16 | **-0.51 ± 0.04** | **-0.80 ± 0.51** | **-0.89 ± 0.52** |
|  | SpaceInvader | 3.87 ± 0.16 | **4.23 ± 0.09** | 2.17 ± 0.04 | **2.36 ± 0.03** | 3.40 ± 0.07 | **3.73 ± 0.12** |
|  | BeamRider | 1.94 ± 0.05 | **2.06 ± 0.03** | 0.75 ± 0.04 | **0.84 ± 0.02** | 0.65 ± 0.01 | **0.69 ± 0.02** |
|  | Seaquest | 5.01 ± 0.30 | **8.50 ± 0.54** | 0.31 ± 0.10 | **1.57 ± 0.03** | 3.13 ± 0.20 | **3.89 ± 0.28** |
|  | MsPacman | 12.45 ± 0.15 | **14.59 ± 0.17** | 10.08 ± 0.25 | **13.85 ± 0.19** | 13.62 ± 0.28 | **14.47 ± 0.23** |
| Num resets | Breakout | 39.80 ± 11.86 | **32.50 ± 2.18** | 84.50 ± 26.40 | **32.70 ± 0.68** | 35.10 ± 0.97 | 52.30 ± 2.97 |
|  | Pong | **4.30 ± 0.21** | 4.90 ± 0.18 | **3.90 ± 0.53** | 4.50 ± 0.17 | **7.80 ± 0.81** | **7.90 ± 0.87** |
|  | SpaceInvader | **12.70 ± 0.76** | 19.30 ± 0.94 | **10.30 ± 0.63** | 32.30 ± 1.05 | **36.60 ± 1.05** | 37.70 ± 0.86 |
|  | BeamRider | **5.10 ± 0.87** | 15.00 ± 1.41 | **3.10 ± 0.59** | 9.40 ± 0.75 | 18.70 ± 0.76 | 20.10 ± 0.69 |
|  | Seaquest | **10.60 ± 0.60** | 13.70 ± 1.52 | **3.30 ± 1.24** | 38.00 ± 1.22 | 18.20 ± 0.57 | 17.20 ± 0.20 |
|  | MsPacman | **32.80 ± 0.55** | 34.90 ± 1.14 | **31.30 ± 0.70** | 34.90 ± 0.53 | **33.30 ± 0.78** | 34.70 ± 0.84 |

Table 13: A comparison of policies learned in the continuing Atari testbeds versus policies learned in the corresponding episodic testbeds. The upper group shows the mean and the standard error of the reward rates when deploying the learned policy obtained in these two settings for $10,000$ steps. The higher reward rate is marked in boldface, and the number obtained in other settings is also marked in bold if the difference is statistically insignificant. The lower group shows the number of resets within the evaluation steps, with the fewer number of resets indicated in bold. This table shows that policies learned in continuing testbeds make more frequent resets and achieve a higher reward rate.

| | Task | DDPG | | TD3 | | SAC | | PPO | |
|---|---|---|---|---|---|---|---|---|---|
| | | agent-controlled | predefined | agent-controlled | predefined | agent-controlled | predefined | agent-controlled | predefined |
| Reward rate | HalfCheetah | **13.15 ± 0.20** | 12.19 ± 1.41 | **12.03 ± 0.43** | 10.48 ± 1.69 | 12.69 ± 0.28 | **14.23 ± 0.77** | **6.81 ± 0.49** | 3.04 ± 0.74 |
| | Ant | **7.32 ± 0.14** | 6.79 ± 0.37 | 5.97 ± 0.25 | **6.78 ± 0.09** | 6.90 ± 0.14 | **7.58 ± 0.20** | **4.41 ± 0.37** | 3.61 ± 0.47 |
| | Hopper | **4.09 ± 0.17** | **4.05 ± 0.06** | **4.34 ± 0.06** | 4.07 ± 0.04 | **4.05 ± 0.09** | **4.19 ± 0.07** | 3.55 ± 0.16 | **4.02 ± 0.07** |
| | Humanoid | 4.10 ± 1.30 | **6.50 ± 0.60** | 5.38 ± 0.04 | **7.75 ± 0.44** | 5.88 ± 0.08 | **8.09 ± 0.09** | 6.82 ± 0.06 | **7.65 ± 0.08** |
| | Walker2d | 4.30 ± 0.14 | **4.88 ± 0.19** | 4.16 ± 0.18 | **4.37 ± 0.50** | **4.29 ± 0.29** | **4.06 ± 0.83** | **4.56 ± 0.25** | **4.87 ± 0.29** |
| Num resets | HalfCheetah | **1.40 ± 0.34** | **1.80 ± 0.96** | **1.20 ± 0.25** | 7.50 ± 5.67 | 6.50 ± 1.28 | **0.70 ± 0.30** | 6.00 ± 1.37 | **0.40 ± 0.31** |
| | Ant | **7.10 ± 1.87** | 23.00 ± 2.67 | 10.20 ± 1.17 | **1.20 ± 0.29** | **2.30 ± 0.63** | 5.70 ± 2.95 | 66.60 ± 24.96 | **4.50 ± 1.52** |
| | Hopper | **37.30 ± 3.23** | 45.50 ± 1.92 | **41.00 ± 1.54** | 45.90 ± 1.60 | **47.70 ± 1.63** | 46.90 ± 2.42 | 56.50 ± 3.37 | **52.90 ± 1.88** |
| | Humanoid | 1102.90 ± 988.81 | **228.10 ± 75.23** | 255.10 ± 7.46 | **55.30 ± 20.32** | 166.80 ± 14.14 | **5.50 ± 1.93** | 125.00 ± 5.95 | **107.40 ± 3.96** |
| | Walker2d | 74.70 ± 6.92 | **42.50 ± 11.94** | 64.30 ± 26.00 | **35.30 ± 15.76** | **91.00 ± 25.29** | 103.70 ± 69.34 | 31.70 ± 5.66 | **28.70 ± 6.15** |

Table 14: A comparison of policies learned in testbeds with predefined resets versus those learned in testbeds with agent-controlled resets. The upper group shows the mean and the standard error of the reward rates when deploying learned policies obtained in these two settings for $10,000$ steps. The higher reward rate is highlighted in bold, and if the difference is statistically insignificant, both values are also marked in bold. The lower group shows the number of resets within the evaluation steps, with the fewer number of resets indicated in bold. In general, algorithms achieve a higher reward rate and lower reset frequency when running on testbeds with predefined resets compared to those where resets are controlled by the agent.

# C   ADDITIONAL RESULTS OF ALGORITHMS WITH REWARD CENTERING

This appendix presents results concerning reward centering that are omitted in the main text. We will start with the result showing the usefulness of TD-based reward centering when a large discount factor or a large reward offset is present. We will then compare the three reward-centering approaches detailed in Section A.4.

To examine the influence of the discount factor with TD-based reward centering, we show the percentage of improvement of the asymptotic reward rate when using a discount factor of 0.999 and 1.0, as compared to when using the discount factor of 0.99, for centered algorithms in Mujoco testbed. Section 2.4 shows the formal definition of this percentage of improvement. As a baseline, we show the percentage of improvement when using a discount factor of 0.999, as compared to a discount factor of 0.99, for the based algorithms (c.f. Table 4). Discount factor 1.0 is not used by the base algorithms because approximate values can diverge to infinity. The results shown in Table 15 suggest that centered algorithms are indeed significantly less sensitive to the choice of the discount factor when resets are available. In several testbeds without resets, including Swimmer, HumanoidStandup, and SpecialAnt, increasing the discount factor can hurt performance significantly. This is potentially due to the fact that none of the tested algorithms, regardless of whether they use reward centering or not, successfully solve these testbeds even with a discount factor of 0.99. A larger discount factor increases the difficulty by optimizing a longer-term value, resulting in even worse performance of the tested algorithms.

To examine the influence of reward offsets, we show the percentage of improvement of the asymptotic reward rate when shifting all rewards by -100 or +100 for both centered and base algorithms in the Mujoco testbeds. Section 2.4 shows the formal definition of this percentage of improvement. The results (Table 16) show that centered algorithms are not sensitive to reward offsets at all, while uncentered algorithms are extremely sensitive to the offsets.

Overall, our experiments confirm the effectiveness of TD-based reward centering by showing that it can be combined with all tested algorithms and improve their performance. Further, centered algorithms work well with a large discount factor, especially for testbeds with resets, making the selection of an appropriate discount factor easier. Finally, the centered algorithm is not sensitive to reward offsets at all.

| | Algorithm | DDPG | | | TD3 | | | SAC | | | PPO | | |
|---|---|---|---|---|---|---|---|---|---|---|---|---|---|
| | Use TD-based RC | Y | Y | N | Y | Y | N | Y | Y | N | Y | Y | N |
| | Discount factor | 0.999 | 1.0 | 0.999 | 0.999 | 1.0 | 0.999 | 0.999 | 1.0 | 0.999 | 0.999 | 1.0 | 0.999 |
| No resets | Swimmer | -88.54 | -96.33 | -85.95 | 71.53 | 75.63 | -45.19 | -52.03 | -95.04 | -99.23 | 102.98 | 122.57 | 46.84 |
| | HumanoidStandup | -11.37 | -38.71 | -9.09 | -5.49 | -11.43 | 14.16 | -46.96 | -52.99 | -60.13 | -9.10 | -13.72 | -13.45 |
| | Reacher | -62.35 | -56.17 | -707.42 | -1.88 | -0.82 | -6.01 | -1.28 | 0.48 | -10.13 | -1.35 | -1.42 | 1.60 |
| | Pusher | -3.79 | -4.05 | -13.80 | -2.83 | -3.34 | -10.82 | -2.93 | -2.78 | -7.23 | -3.39 | -4.06 | -4.54 |
| | SpecialAnt | 32.19 | -9.86 | -38.39 | -38.02 | -56.02 | -67.71 | -153.05 | -148.25 | -152.16 | -6.75 | -23.00 | -11.86 |
| Predefined resets | HalfCheetah | -10.82 | 15.43 | -20.62 | 20.16 | 15.41 | 49.84 | -6.24 | -6.06 | 4.26 | 7.59 | -19.77 | -41.32 |
| | Ant | -2.84 | -0.57 | -7.48 | -2.31 | -2.98 | -22.46 | -2.50 | 0.75 | -14.66 | 9.33 | -6.29 | -15.50 |
| | Hopper | -2.18 | -2.07 | -12.81 | 0.93 | 0.53 | -8.45 | -0.65 | -6.78 | -11.72 | -1.46 | -6.20 | -21.73 |
| | Humanoid | -8.19 | -1.58 | -34.28 | -29.44 | -30.60 | -64.27 | -14.40 | -14.99 | -74.81 | -6.10 | -2.34 | -58.51 |
| | Walker2d | 0.25 | 1.35 | -5.07 | -15.93 | -13.34 | -15.12 | -4.25 | -9.44 | -3.38 | -5.25 | -10.15 | -29.89 |
| Agent-controlled resets | HalfCheetah | 2.28 | -8.50 | -27.19 | -4.49 | -4.07 | -29.79 | -6.20 | -2.15 | -33.93 | 1.10 | 9.42 | -26.41 |
| | Ant | 1.06 | -1.55 | -5.32 | 4.15 | 2.95 | 10.74 | -4.76 | -20.34 | -22.89 | -11.63 | -6.85 | -19.04 |
| | Hopper | -9.09 | -26.14 | -29.62 | 0.54 | 0.47 | -11.62 | 2.85 | 3.84 | -5.92 | -0.29 | -2.23 | -12.87 |
| | Humanoid | -37.50 | -86.99 | -155.59 | 3.60 | -0.17 | 2.13 | -1.66 | -3.29 | -14.77 | -1.53 | -3.27 | -23.05 |
| | Walker2d | -17.47 | -23.27 | -30.49 | -11.93 | -7.72 | 6.14 | -5.48 | -22.19 | -37.61 | -13.20 | -12.19 | -22.96 |

Table 15: TD-based reward centering is less sensitive to the choice of the discount factor than noncentered methods

We now compare the three reward-centering methods. Tables 17 and 18 display the percentage of improvement for each method, with results for the TD-based approach drawn from Tables 5 and 6. The experimental setup and method for calculating the percentage improvement are detailed in Section 3, while hyperparameters specific to each reward-centering method are provided in Section A.4. We show the learning curves of the base algorithms and the algorithms with various reward-centering approaches in Figures 4—7.

The results indicate that the TD-based approach performs best among the three methods tested. Interestingly, the moving-average approach also performed well in off-policy algorithms, which was unexpected. The reference-state-based approach showed mixed results, improving performance in some cases but diminishing it in others.

| Algorithm | DDPG | | | | TD3 | | | | SAC | | | | PPO | | | |
|---|---|---|---|---|---|---|---|---|---|---|---|---|---|---|---|---|
| Reward shifting | -100 | | +100 | | -100 | | +100 | | -100 | | +100 | | -100 | | +100 | |
| Use TD-based RC | Y | N | Y | N | Y | N | Y | N | Y | N | Y | N | Y | N | Y | N |
| **No resets** Swimmer | 32.03 | -113.27 | -3.89 | -104.86 | -12.03 | -103.63 | -3.35 | -103.20 | -45.95 | -117.02 | -32.34 | -108.30 | 9.90 | -103.27 | 5.42 | -101.52 |
| HumanoidStandup | -14.40 | 44.69 | -14.32 | -29.16 | -18.89 | 41.30 | -6.92 | 5.70 | 19.61 | -32.88 | 9.26 | -24.10 | 10.76 | -1.48 | -0.03 | -11.97 |
| Reacher | 5.86 | -295.90 | 1.75 | -429.94 | 1.77 | -112.49 | 1.64 | -160.87 | 0.44 | -32.00 | 0.37 | -117.87 | 0.10 | -3.09 | -0.28 | -8.67 |
| Pusher | 1.16 | -73.46 | -0.63 | -183.44 | 1.48 | -86.38 | 0.20 | -162.26 | 0.26 | -20.45 | -1.29 | -25.07 | -1.75 | -10.79 | -1.47 | -19.53 |
| SpecialAnt | -11.15 | -156.21 | -12.52 | -100.50 | -8.91 | -65.57 | -9.20 | -44.65 | 45.39 | -12.97 | 48.23 | -12.73 | -2.72 | -40.94 | -4.95 | -42.30 |
| **Predefined resets** HalfCheetah | 14.53 | -31.38 | 4.67 | -59.69 | -6.82 | -10.51 | 7.35 | -85.34 | 8.83 | -45.64 | 3.54 | -44.86 | 0.62 | -45.83 | -0.71 | -62.95 |
| Ant | 1.76 | -93.29 | -0.99 | -118.93 | -2.23 | -73.80 | -1.96 | -97.01 | -0.34 | -46.73 | 1.26 | -75.70 | 2.09 | -34.96 | 6.04 | -31.90 |
| Hopper | -0.26 | -41.21 | 0.49 | -62.35 | 0.16 | -49.28 | 2.83 | -53.05 | -2.09 | -28.59 | -2.05 | -17.77 | -3.06 | -44.08 | 0.26 | -36.00 |
| Humanoid | 0.60 | -69.68 | -0.35 | -83.17 | -8.97 | -87.61 | -10.87 | -113.79 | -1.70 | -79.30 | -0.43 | -109.34 | -0.60 | -44.44 | -1.74 | -50.57 |
| Walker2d | -0.66 | -26.45 | -0.41 | -63.41 | -3.15 | -32.44 | -7.01 | -53.33 | -3.01 | -25.64 | -2.07 | -40.23 | -2.54 | -56.70 | 0.24 | -52.50 |
| **Agent-controlled resets** HalfCheetah | 4.33 | -29.17 | 5.82 | -73.96 | 1.13 | -34.89 | -3.03 | -26.31 | -2.78 | -46.60 | -4.77 | -59.21 | -0.64 | -78.67 | 13.89 | -78.05 |
| Ant | 0.59 | -81.13 | 2.41 | -127.31 | 2.93 | -75.54 | 3.49 | -85.00 | 0.09 | -49.99 | -1.52 | -82.81 | -2.20 | -71.29 | -0.72 | -69.68 |
| Hopper | 2.54 | -23.70 | -0.60 | -106.48 | -2.55 | -34.18 | -1.81 | -65.56 | 2.45 | -21.67 | 3.20 | -11.30 | -0.75 | -69.86 | 0.85 | -36.35 |
| Humanoid | -39.22 | -180.58 | -9.35 | -106.27 | 2.30 | -118.38 | -2.99 | -108.54 | 2.22 | -15.29 | 5.33 | -35.35 | -0.99 | -15.79 | 0.93 | -21.05 |
| Walker2d | -4.87 | -41.19 | -0.88 | -59.64 | -11.85 | -6.78 | 3.81 | -46.72 | -1.10 | -31.97 | 6.78 | -35.39 | 2.16 | -72.69 | -2.23 | -77.86 |

Table 16: TD-based reward centering is not sensitive to reward shifting.

It is worth noting that we tested only the simplest choice for the $f$ function—using the mean of action values from a fixed batch of state-action pairs. Other choices for the $f$ function could potentially yield better results. Additionally, recent work has proposed estimating the reward rate by applying a moving average to the $f$ function's output (Hisaki and Ono, 2024). Future research is needed to evaluate alternative choices for the $f$ function and evaluate the moving-average approach within the reference-state-based framework.

| Algorithm | DDPG | | | TD3 | | | SAC | | | PPO | | |
|---|---|---|---|---|---|---|---|---|---|---|---|---|
| RC approach | TD | RVI | MA | TD | RVI | MA | TD | RVI | MA | TD | RVI | MA |
| **No resets** Swimmer | 109.11 | 66.96 | 176.30 | 90.71 | -59.05 | -22.94 | 1149.26 | 809.21 | 481.01 | 71.14 | -86.18 | 34.98 |
| HumanoidStandup | 41.67 | 29.68 | 28.42 | 19.79 | 14.47 | 16.46 | 35.83 | -16.14 | 7.40 | 19.39 | 5.30 | 26.30 |
| Reacher | -0.03 | -74.41 | -0.01 | 0.07 | -78.37 | 0.03 | -0.11 | -0.14 | -0.08 | 1.17 | 1.26 | 1.20 |
| Pusher | 10.87 | -16.80 | 10.70 | 1.24 | -17.14 | 1.00 | 0.39 | -2.58 | -0.20 | 3.72 | 0.09 | 2.92 |
| SpecialAnt | 12.67 | -4.02 | 21.23 | 2.05 | -9.34 | -3.01 | 5.59 | -18.81 | 5.15 | 10.55 | 9.59 | 4.38 |
| **Predefined resets** HalfCheetah | 3.15 | 0.25 | 4.07 | 13.13 | 5.24 | 4.65 | 5.05 | 1.37 | 0.43 | 4.66 | -0.14 | 12.69 |
| Ant | 22.22 | -14.27 | 13.10 | 18.25 | -19.80 | 23.04 | 6.75 | 6.12 | 9.33 | 13.36 | 4.32 | 1.36 |
| Hopper | 2.53 | -20.79 | 1.52 | 14.83 | -16.84 | 15.37 | 4.44 | 3.83 | 4.13 | 4.56 | -2.31 | 3.23 |
| Humanoid | 210.54 | 163.26 | 213.03 | 89.68 | 52.56 | 70.91 | 77.54 | 61.66 | 51.39 | 11.29 | 16.17 | 9.27 |
| Walker2d | 16.28 | 0.20 | 7.87 | 10.37 | -11.10 | 8.64 | 7.72 | 6.79 | 6.22 | 7.37 | 5.44 | 15.34 |
| **Agent-controlled resets** HalfCheetah | 2.21 | 1.89 | 1.53 | 17.45 | 19.86 | 13.88 | -2.05 | -9.81 | -18.64 | 4.06 | 13.00 | 30.94 |
| Ant | 12.73 | 12.90 | 3.99 | 94.13 | 58.32 | 104.65 | 34.57 | 30.92 | 26.04 | 8.07 | -8.31 | -4.47 |
| Hopper | 42.28 | 17.31 | 30.28 | 15.72 | 16.55 | 8.12 | 4.60 | -3.18 | 5.04 | 5.57 | 2.87 | 7.26 |
| Humanoid | 246.73 | 126.40 | 115.99 | 20.71 | 14.63 | 23.19 | 5.46 | 4.01 | -3.28 | 2.36 | 1.98 | 0.32 |
| Walker2d | 10.23 | -7.59 | -0.10 | 12.92 | -10.86 | 0.79 | 0.89 | -17.88 | -5.99 | 4.61 | 0.19 | 7.29 |
| Average improvement | 49.54 | 18.73 | 41.86 | 28.07 | -2.72 | 17.65 | 89.06 | 57.02 | 37.86 | 11.46 | -2.45 | 10.20 |

Table 17: Performance improvement when applying reward centering to the tested algorithms to solve the Mujoco testbed. Here, RVI standards for the reference-state-based approach. MA standards for the moving-average-based approach.

| Task | DQN | | | SAC | | | PPO | | |
|---|---|---|---|---|---|---|---|---|---|
| RC approach | TD | RVI | MA | TD | RVI | MA | TD | RVI | MA |
| Breakout | -7.48 | -31.97 | -6.79 | 1.67 | 2.10 | -0.22 | 11.51 | 0.73 | 3.12 |
| Pong | 0.50 | -33.64 | 2.45 | 51.94 | 50.46 | -0.13 | 79.18 | 30.13 | 68.90 |
| SpaceInvader | 20.97 | 15.93 | 12.93 | 0.29 | -2.54 | 0.73 | 19.72 | 35.18 | 7.96 |
| BeamRider | 7.01 | 6.82 | 3.59 | 35.00 | 13.05 | 0.67 | 75.67 | 4.88 | 72.31 |
| Seaquest | 26.79 | -4.38 | 19.42 | 22.75 | -10.35 | 0.24 | 5.77 | -12.69 | -4.65 |
| MsPacman | 9.96 | 8.70 | 4.58 | 1.76 | -0.42 | 1.50 | 2.67 | -0.61 | 2.41 |
| Average improvement | 9.63 | -6.42 | 6.03 | 18.90 | 8.72 | 0.465 | 32.42 | 9.60 | 25.01 |

Table 18: The performance improvement when applying reward centering in the tested algorithms to solve the Atari testbeds.

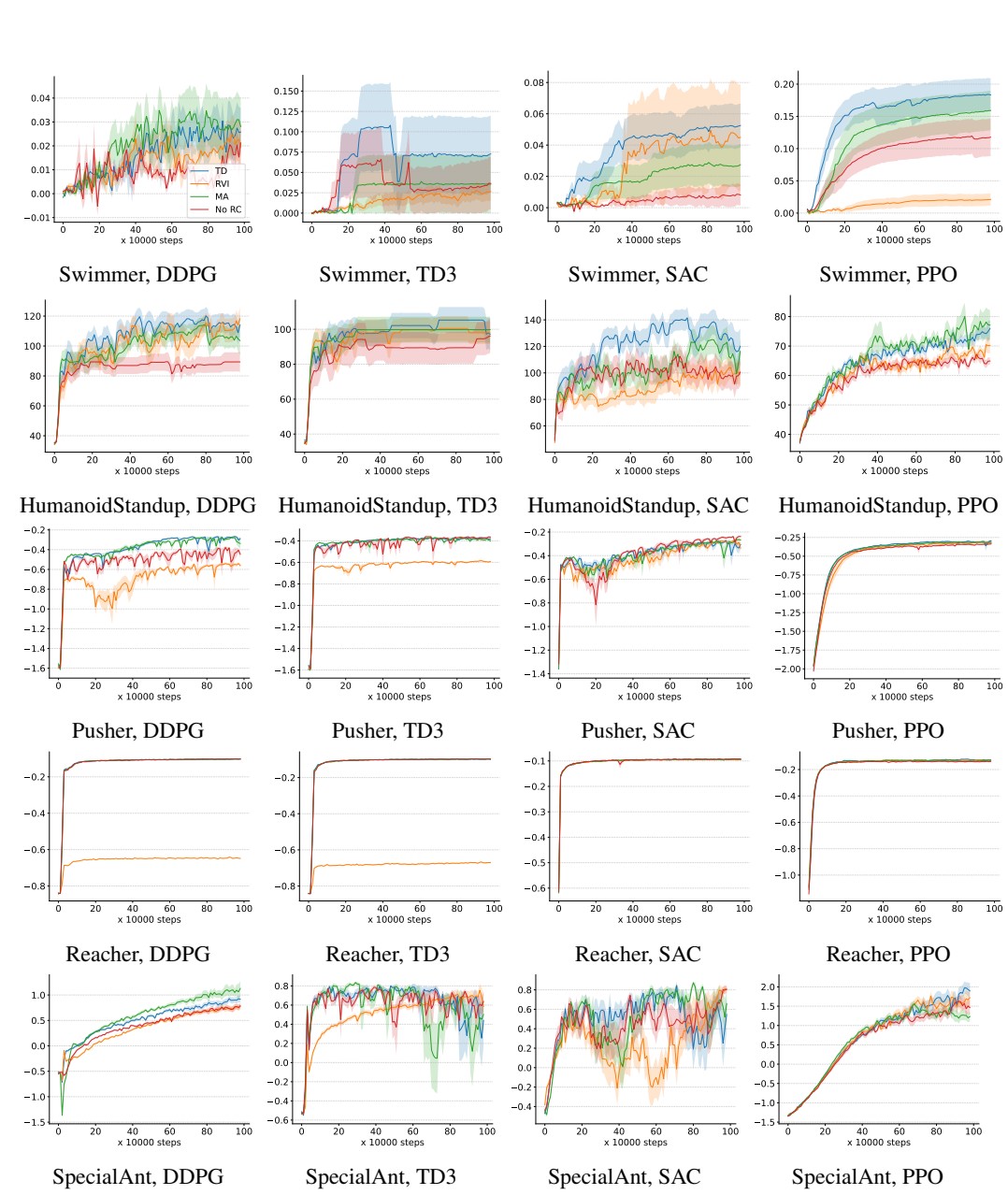

Figure 4: Learning curves on continuing testbeds without resets based on Mujoco environments. Each point shows the reward rate averaged over the past 10,000 steps.

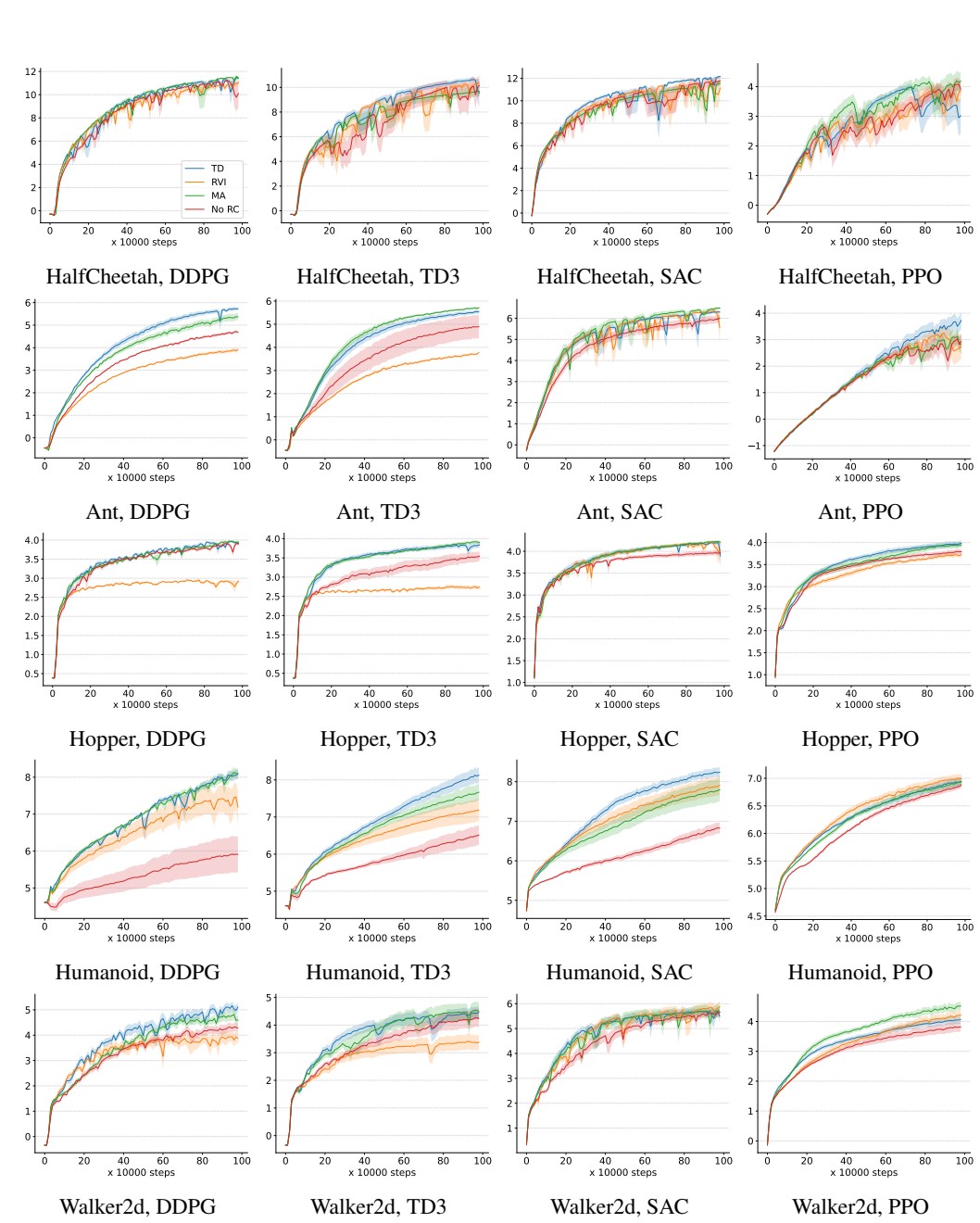

Figure 5: Learning curves on continuing testbeds with predefined resets based on Mujoco environments. Each point shows the reward rate averaged over the past 10,000 steps. The shading area standards for one standard error.

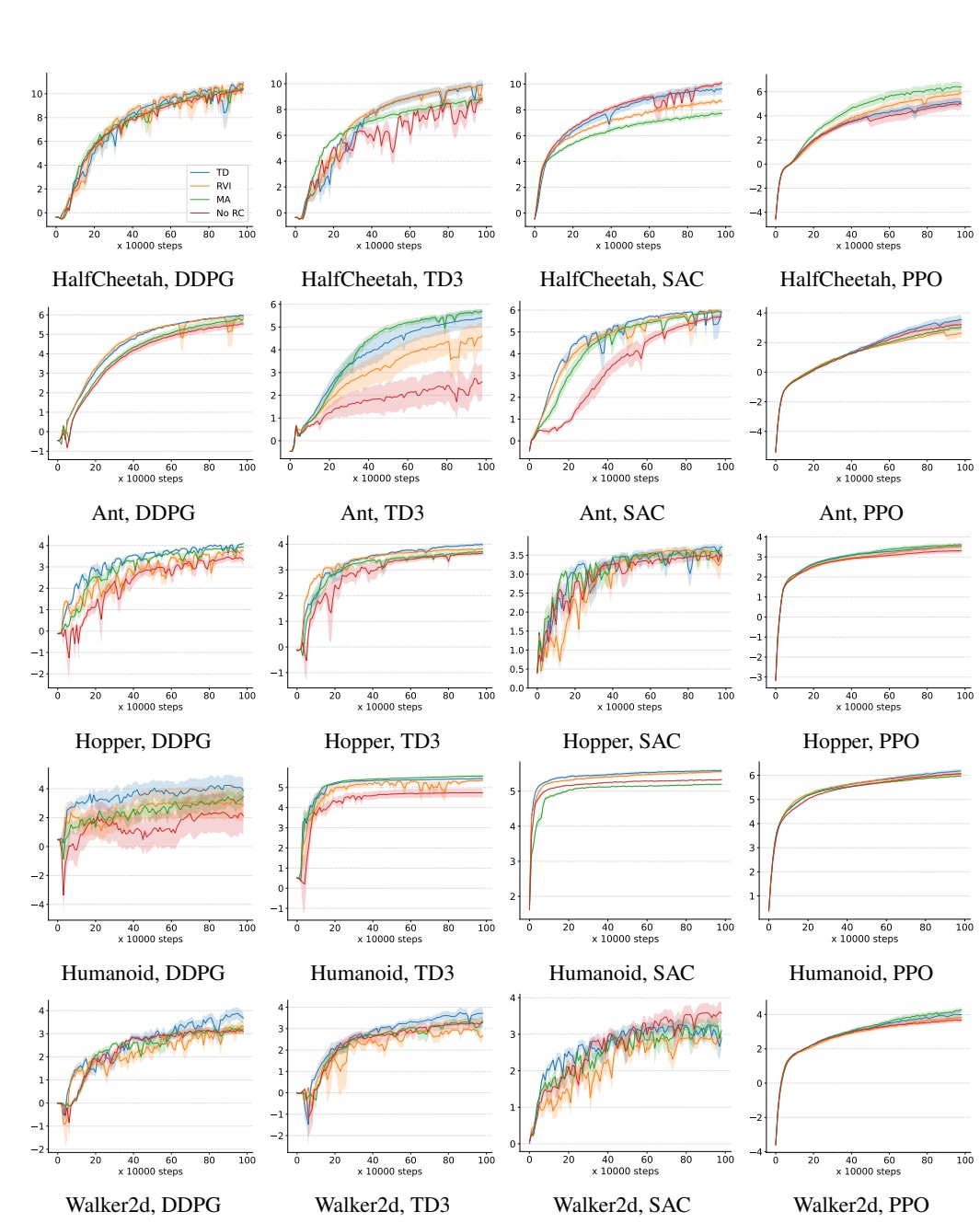

Figure 6: Learning curves on continuing testbeds with agent-controlled resets based on Mujoco environments. Each point shows the reward rate averaged over the past 10,000 steps. The shading area standards for one standard error.

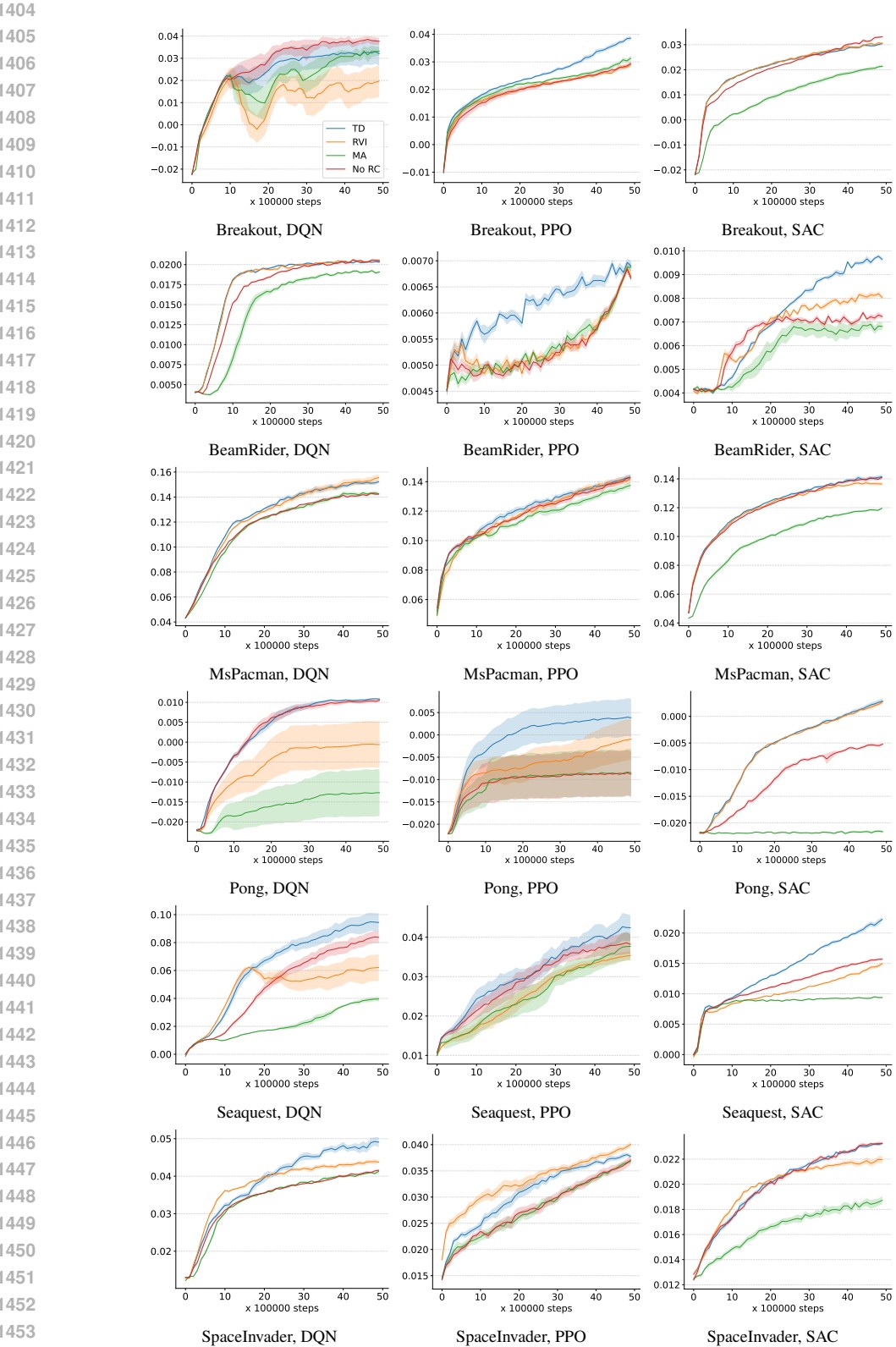

Figure 7: Learning curves on continuing testbeds with predefined resets based on Atari environments. Each point shows the reward rate averaged over the past 10,000 steps. Shading area standards for one standard error.