# OpenReview forum: "An Empirical Study of Deep Reinforcement Learning in Continuing Tasks"
_ICLR.cc/2025/Conference — ICLR 2025 Conference Withdrawn Submission_

### Official Review · Reviewer_eWSU · 2024-10-28

**Soundness:** 3
**Presentation:** 1
**Contribution:** 2
**Rating:** 3
**Confidence:** 4

**Summary:**

This work presents an empirical study of DRL in continuing tasks. To evaluate the performance of typical DRL agents in continuing tasks, three types of continuing testbeds are proposed through modifying MuJoCo and Atari tasks, including testbeds without resets, testbeds with predefined resets, and testbeds with agent-controlled resets. With experiments conducted on these testbeds, this work shows the ineffectiveness and learning issues of typical DRL agents in continuing tasks especially when larger discount factors or reward offsets are used. Moreover, the experiments demonstrate the effectiveness of reward centering approach in improving learning performance in continuing tasks.

**Strengths:**

- To my knowledge, learning in continuing tasks is of great practical significance while remaining under-explored in the RL community. This work proposes a promising standard benchmark for this direction.
- For the experiments, three types of testbeds are proposed and popular DRL agents are used as baselines. A visual analysis of the failure patterns is provided in Figure 2.

**Weaknesses:**

- The writing is unsatisfactory, especially I believe there is a large room to improve the organization of the content. Since this work is mainly on presenting an empirical study, the purposes and structures of experiment designs and the key findings of experimental results are of the most importance to the audience. Here are a few suggestions:
    - The introduction section is a bit too long. It is now over 2 pages, which makes it difficult to convey the main thread of thoughts and findings in a clear way. A suggestion here is to add a Related Work section.
    - In Section 2, three types of testbeds are proposed one by one with a long subsection for each. My suggestion here is to add an overall description of the experiment designs and the target questions to investigate at the beginning of Section 2, then navigate to each subsection, which I think will help the reading and understanding of the audience greatly.
    - The experimental results on Atari tasks are not well discussed in the main body of the paper. I suggest the author adjust the content and include them.
- The continuing tasks considered in this work are from popular typical RL benchmarks but are still lacking in representativity. MuJoCo tasks are basically locomotion tasks, except for pusher and reacher are goal-reaching tasks, where the optimal control has a periodical feature intuitively. This cripples the significance of continuing learning to some degree. As mentioned by the authors, real-world problems like inventory management, content recommendation, and portfolio management, are story-like and indeed Atari tasks should be better options for building continuing tasks. Fortunately, the authors include 6 Atari tasks in this work but more emphasis is put on MuJoCo tasks. In addition, Minecraft could be a good choice for continuing tasks. It will be great if they are included in the future.
- For Section 2.3, it would be great to see an analysis of the learned reset control by the agents.
- In Figure 2, two concrete failure cases of learning in continuing tasks without resets are presented. However, I feel a bit lacking in further discussion on the two reasons.

&nbsp;

### Minors

- The margin below Table 6 is too narrow.

**Questions:**

1. For Table 2 and 3, the numbers of resets for DDPG in Ant with the reset cost 1/10/100 are 94.20 ± 5.98, 23.00 ± 2.67, 65.20 ± 4.76, similarly, the numbers of resets for TD3 in Ant with the reset cost 1/10/100 are 89.80 ± 10.27, 1.20 ± 0.29, 58.40 ± 9.65. It seems make no sense. I was wondering if there were some mistakes in these numbers?
2. Also for Table 3, the numbers of resets for DDPG in Walker2d with the reset cost 1/100 are 104.50 ± 17.33, 39.70 ± 3.98; while the corresponding reward rates are 3.79 ± 0.14, 3.95 ± 0.11. It looks like the number of resets does not influence the reward rate much. Is there any explanation or further analysis for this? By the way, this is not the only place showing such a confusion. I suggest the authors re-check the numbers reported in Table 2 and 3 in case there are some mistakes.

---

### Official Review · Reviewer_yYkn · 2024-10-30

**Soundness:** 3
**Presentation:** 3
**Contribution:** 3
**Rating:** 6
**Confidence:** 4

**Summary:**

This work presents an extensive analysis of the impact of resets on the performance of the agent (as measured by reward rates).
There are some interesting insights and findings that may lead to improved algorithm and environment design in the future.

**Strengths:**

The experimentation is well done.
There is a good analysis under the reward-rate metric.
Diving into WHY this happens and then drawing conclusions for Swimmer and this leading to dramatic improvements is very impressive.

**Weaknesses:**

The whole premise of the work lies on the connection to real world tasks, primarily robotics.
When simulation is not possible (for example, in very complex scenarios), then it is indeed of interest to deploy the robot in the real world and have it continually learn.

However, I feel that there is some mixture between theory and practice, specifically around desired metrics to measure.
When deploying a robot, I don't see the reward rate being a metric of interest. There's a task, we can measure how well it is on the task. Even if it is trained in a continual setting and not a classic episodic one.

**Questions:**

I would have liked to see the actual task metrics and how they compare.
Reward rate correlates with training, but how would it perform at inference once deployed.

For example -- multiple Reacher agents are trained with the various resetting schemes.
Then, at inference, they are all measured on the standard non-reset episodic setting.
In the end, for most robotic tasks there is a clear success and error metric that can be measured, and tasks have a clear end.

---

### Official Review · Reviewer_BqsR · 2024-10-30

**Soundness:** 2
**Presentation:** 1
**Contribution:** 2
**Rating:** 3
**Confidence:** 4

**Summary:**

This paper evaluates the performance of standard deep RL algorithms in the continuing task setting. The paper makes two contributions. First, it reports findings from an empirical study on MuJoCu environments subject to three reset scenarios (no resets, predefined resets, and agent-controlled resets). The authors find that performance is considerably worse without resets that with them. Second, they propose fixes inspired by Naik et al.'s (2024) _reward centering_, which hasn't yet been applied in the deep RL setting, and show it improves performance.

**Strengths:**

- The experimental setup is, to the best of my knowledge, novel and interesting. RL in the contuining task setting is an important sub-area, with many practical applications.
- The empirical study in Section 2 is extensive; they implement all of the canonical deep RL algorithms, and perform enough runs (10 per agent/environment) to establish statistical significance.

**Weaknesses:**

- The core methodological proposals (Section 3) represent what is, in my opinion, an incremental amendment to Naik et al.'s reward centering work. The authors show that Naik et al.'s proposals can be repurposed for the deep RL setting, but this is a logical conclusion one could draw from Naik et al.'s paper alone, and does not represent the kind of methodological novelty usually expected from a paper at ICLR.
- The paper's layout feels unintuitive. Though the authors discuss some related work in the introduction, the paper lacks a dedicated related work section in which they clearly situate their contributions in the context of others'. Equally, the authors do not formally introduce their problem and its associated notation in the main body. This becomes particularly problematic in Section 3 when they start building upon Naik et al.'s notation without prior discussion (I appreciate the authors introduce MDPs etc. in Appendix A.1, but this should come prior to the Appendix). I recommend that the authors stick to standard convention and follow something akin to: Introduction -> Related Work -> Preliminaries / Background -> Methods -> Results -> Discussion/Limitations -> Conclusions.
- The authors do not justify several decisions in the design of their problem setting. For example, in lines 146-156 they state the target position in Reacher is resampled every 50 steps, but for Pusher it is resampled every 100 steps. And in lines 294-295 they state that "[e]ach reset incurs a penalty of −10 to the reward, punishing the agent for falling or flipping". I suspect agent performance is likely sensitive to these decisions, but they are made with little justification or reference to prior work.
- The authors often use imprecise language to describe their results e.g. "decent policies" (lines 200, 202), "policy that performed reasonably well" (line 205). I would recommend that they stick to quantitative comparisons.
- In Lines 257-259, the authors state that "the agent did not perform sufficient exploration to escape from suboptimal states."  Methods like SAC and TD3 are generally evaluated with the exploration noise turned off (i.e. the mean of the action distribution is used, rather than a sample from the distribution). If the authors followed this convention then this could potentially explain the results, but because they do not discuss their implementations in detail, it is unclear whether that convention was followed.

Some minor feedback:
- In Figure 1 it would be helpful if the y-axis was fixed for each environment so the reader can more easily see how performance varies across reset settings for each environment.
- Gymnasium (Towers et al., 2024) builds upon Gym (Brockman et al., 2016); I'd recommend citing the latter too.
- Spelling of "decent" in line 200
- Missing "the" in line 204

**Questions:**

- How do the methods perform in the standard episodic setting? It would be helpful to report this as an upper-bound which the reader could use to establish performance degradation in each of your continuing task settings.
- Why should the reader be concerned that performance of the evaluated methods is significantly reduced with high discount factors and reward offsets? Are these situations practically relevant? And, following that, what is the theoretical justification for the performance reductions under these conditions?
- How does performance in Section 2 change when exploration hyperparameters are varied? (E.g. the exploration noise distribution for TD3). Can we elicit better performance by increasing them?
- Why does giving the agent control over its own resets reduce performance? Shouldn't the agent be able to learn the predefined resets from Section 2.2?
- In Section 2.1 you select 5 environments that exhibit the weakly communicating MDP property. Why do you then change the evaluation environments in Sections 2.2 and 2.3? Could you not implement predefined resets and agent-controlled resets in the environments from Section 2.1? If you fixed the environments and only varied the reset conditions it would be easier to draw firm conclusions about the effect of resets on agent performance.

---

### Note · Authors · 2024-11-27

**Comment:**

We sincerely thank all the reviewers for their detailed and constructive feedback.

After carefully considering the reviews, we recognize that the primary concerns pertain to the organization of the paper. Upon reflection, we agree that the current structure needs improvements and are actively working on a revised version with significant organizational changes. Given the extent of these changes, we believe it would be inappropriate to request the reviewers to evaluate the revised version. As a result, we have decided to withdraw the current submission.

**Withdrawal Confirmation:**

I have read and agree with the venue's withdrawal policy on behalf of myself and my co-authors.